# ROBUST CLASSIFICATION VIA A SINGLE DIFFUSION MODEL

## ABSTRACT

Recently, diffusion models have been successfully applied to improving adversarial robustness of image classifiers by purifying the adversarial noises or generating realistic data for adversarial training. However, the diffusion-based purification can be evaded by stronger adaptive attacks while adversarial training does not perform well under unseen threats, exhibiting inevitable limitations of these methods. To better harness the expressive power of diffusion models, in this paper we propose Robust Diffusion Classifier (RDC), a generative classifier that is constructed from a pre-trained diffusion model to be adversarially robust. Our method first maximizes the data likelihood of a given input and then predicts the class probabilities of the optimized input using the conditional likelihood estimated by the diffusion model through Bayes' theorem. To further reduce the computational complexity, we propose a new diffusion backbone called multi-head diffusion and develop efficient sampling strategies. As our method does not require training on particular adversarial attacks, we demonstrate that it is more generalizable to defend against multiple unseen threats. In particular, RDC achieves 75.67% robust accuracy against $\ell_\infty$ norm-bounded perturbations with $\epsilon_\infty = 8/255$ on CIFAR-10, surpassing the previous state-of-the-art adversarial training models by $+4.77\%$. The findings highlight the potential of generative classifiers by employing diffusion models for adversarial robustness compared with the commonly studied discriminative classifiers.

## 1 INTRODUCTION

A longstanding problem of deep learning is the vulnerability to adversarial examples (Szegedy et al., 2014; Goodfellow et al., 2015), which are maliciously generated by adding human-imperceptible perturbations to natural examples, but can cause deep learning models to make erroneous predictions. Since the adversarial robustness problem leads to security threats in real-world applications (e.g., face recognition (Sharif et al., 2016; Dong et al., 2019), autonomous driving (Cao et al., 2021; Jing et al., 2021), healthcare (Finlayson et al., 2019)), there has been a lot of work on defending against adversarial examples, such as adversarial training (Goodfellow et al., 2015; Madry et al., 2018; Zhang et al., 2019), image denoising (Liao et al., 2018; Samangouei et al., 2018; Song et al., 2018), certified defenses (Raghunathan et al., 2018; Wong & Kolter, 2018; Cohen et al., 2019).

Recently, diffusion models have emerged as a powerful family of generative models, consisting of a forward diffusion process that gradually perturbs data with Gaussian noises and a reverse generative process that learns to remove noise from the perturbed data (Sohl-Dickstein et al., 2015; Ho et al., 2020; Nichol & Dhariwal, 2021; Song et al., 2021). Some researchers have tried to apply diffusion models to improving adversarial robustness in different ways. For example, the adversarial images can be purified through the forward and reverse processes of diffusion models before feeding into the classifier (Blau et al., 2022; Nie et al., 2022; Wang et al., 2022). Besides, the generated data from diffusion models can significantly improve adversarial training (Rebuffi et al., 2021; Wang et al., 2023a), achieving the state-of-the-art results on robustness benchmarks (Croce et al., 2020). These works show promise of diffusion models in the field of adversarial robustness.

However, there are still some limitations of the existing methods. On one hand, the diffusion-based purification approach is a kind of gradient obfuscation (Athalye et al., 2018; Gao et al., 2022), and can be effectively attacked by using the exact gradient and a proper step size[1]. We observe that the

---

[1]We lower the robust accuracy of DiffPure (Nie et al., 2022) from 71.29% to 53.52% under the $\ell_\infty$ norm with $\epsilon_\infty = 8/255$, and from 80.60% to 75.59% under the $\ell_2$ norm with $\epsilon_2 = 0.5$, as shown in Table 1.

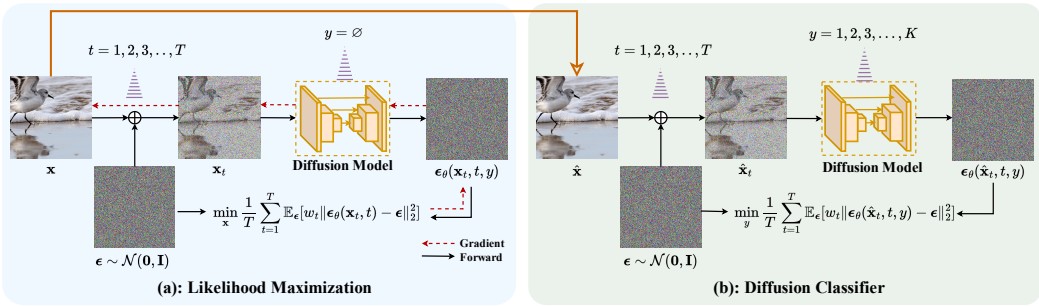

Figure 1: Illustration of our proposed Robust Diffusion Classifier (RDC). Given an input image $\mathbf{x}$, our approach first maximizes the data likelihood (Left) and then classifies the optimized image $\hat{\mathbf{x}}$ with a diffusion model (Right). The class probability $p(y|\hat{\mathbf{x}})$ is given by the conditional log-likelihood $\log p_\theta(\hat{\mathbf{x}}|y)$, which is approximated by the variational lower bound involving calculating the noise prediction error (i.e., diffusion loss) averaged over different timesteps for every class.

adversarial example cannot make the diffusion model output an image of a different class, but the perturbation is not completely removed. Therefore, the poor robustness of diffusion-based purification is largely due to the vulnerability of downstream classifiers. On the other hand, although adversarial training methods using data generated by diffusion models achieve excellent performance, they are usually not generalizable across different threat models (Tramèr & Boneh, 2019). In summary, these methods leverage diffusion models to improve adversarial robustness of discriminative classifiers, but discriminative learning cannot capture the underlying structure of data distribution, making it hard to control the predictions of inputs outside the training distribution (Schott et al., 2019). As a generative approach, diffusion models provide a more accurate estimation of score function (i.e., the gradient of log-density at the data point) across the entire data space (Song & Ermon, 2019; Ho et al., 2020), which also have the potential to provide accurate class probabilities. Therefore, we try to explore *how to convert a diffusion model into a generative classifier for improved adversarial robustness?*

In this paper, we propose **Robust Diffusion Classifier (RDC)**, a generative classifier obtained from a single pre-trained diffusion model to achieve adversarial robustness. Our method calculates the class probability $p(y|\mathbf{x})$ using the conditional likelihood $p_\theta(\mathbf{x}|y)$ estimated by a diffusion model through Bayes' theorem. The conditional likelihood is approximated by the variational lower bound, which involves calculating the noise prediction loss for every class under different noise levels. In order to avoid time complexity induced by the number of classes, we propose a new UNet backbone named **multi-head diffusion** by modifying the last convolutional layer to output noise predictions of all classes simultaneously. Theoretically, we validate that the optimal diffusion model can achieve absolute robustness under common threat models. However, the practical diffusion model may have an inaccurate density estimation $p_\theta(\mathbf{x}|y)$ or a large gap between the likelihood and its lower bound, leading to inferior performance. To address this issue, we further propose **Likelihood Maximization** as a pre-optimization step to move the input data to regions of high likelihoods before feeding into the diffusion classifier. Our RDC, directly constructed from a pre-trained diffusion model without training on specific adversarial attacks, can perform robust classification under various threat models.

We empirically compare our proposed method with various state-of-the-art methods against strong adaptive attacks. Specifically, on CIFAR-10 (Krizhevsky & Hinton, 2009), RDC achieves 75.67% robust accuracy under the $\ell_\infty$ norm threat model with $\epsilon_\infty = 8/255$, exhibiting a $+4.77\%$ improvement over the state-of-the-art adversarial training method (Wang et al., 2023a), and a $+3.01\%$ improvement over the state-of-the-art dynamic defenses and randomized defenses (Pérez et al., 2021; Blau et al., 2023). Under unseen threats, RDC leads to a more significant improvement ($> 30\%$) over adversarial training models, DiffPure (Nie et al., 2022) and generative classifiers. We further conduct thorough analysis of more carefully-designed adaptive attacks and various ablation studies to verify that the achieved robustness is not caused by gradient obfuscation (Athalye et al., 2018). Our findings disclose the potential of generative models for solving the adversarial robustness problem.

## 2 RELATED WORK

**Adversarial robustness.** Adversarial examples (Szegedy et al., 2014; Goodfellow et al., 2015) are widely studied in the literature, which are generated by adding imperceptible perturbations to

natural examples, but can mislead deep learning models. Many adversarial attack methods (Carlini & Wagner, 2017; Athalye et al., 2018; Dong et al., 2018; Madry et al., 2018; Croce & Hein, 2020) have been proposed to improve the attack success rate under the white-box or black-box settings, which can be used to evaluate model robustness. To defend against adversarial attacks, adversarial training (Madry et al., 2018; Zhang et al., 2019; Rebuffi et al., 2021) stands out as the most effective method, which trains neural networks using adversarially augmented data. However, these models tend to exhibit robustness only to a specific attack they are trained with, and have poor generalization ability to unseen threats (Tramèr & Boneh, 2019; Laidlaw et al., 2021). Another popular approach is adversarial purification (Liao et al., 2018; Samangouei et al., 2018; Song et al., 2018; Nie et al., 2022), which denoises the input images for classification. Most of these defenses cause obfuscated gradients (Athalye et al., 2018) and can be evaded by adaptive attacks (Tramer et al., 2020).

**Generative classifiers.** Generative classifiers, like naive Bayes (Ng & Jordan, 2001), predict the class probabilities $p(y|\mathbf{x})$ for a given input $\mathbf{x}$ by modeling the data likelihood $p(\mathbf{x}|y)$ using generative models. Compared with discriminative classifiers, generative classifiers are often more robust and well-calibrated (Raina et al., 2003; Schott et al., 2019; Li et al., 2019; Mackowiak et al., 2021). Modern generative models like diffusion models (Ho et al., 2020) and energy-based models (LeCun et al., 2006) can also be used as generative classifiers. SBGC (Zimmermann et al., 2021) utilizes a score-based model to calculate the log-likelihood $\log p(\mathbf{x}|y)$ by integration and then calculates $p(y|\mathbf{x})$ via Bayes' theorem. HybViT (Yang et al., 2022) directly learns the joint likelihood $\log p(\mathbf{x}, y) = \log p(\mathbf{x}) + \log p(y|\mathbf{x})$ by training a diffusion model to learn $\log p(\mathbf{x})$ and a standard classifier to model $\log p(y|\mathbf{x})$ at training time, and direct predicts $p(y|\mathbf{x})$ at test time. JEM (Grathwohl et al., 2019) utilizes the energy-based model to predict joint likelihood $\log p(\mathbf{x}, y)$ and Bayes' theorem to get $p(y|\mathbf{x})$. We also compare with these generative classifiers in the experiments. Recently, diffusion models has also been used for generative classification. Hoogeboom et al. (2021); Han et al. (2022) perform diffusion process in logit space to learn the categorial classification distribution. Two concurrent works (Clark & Jaini, 2023; Li et al., 2023) also convert diffusion models to generative classifiers in a similar way to ours, but they mainly focus on zero-shot classification while do not consider the adversarial robustness.

**Diffusion models for adversarial robustness.** As a powerful family of generative models (Dhariwal & Nichol, 2021), diffusion models have been introduced to further improve adversarial robustness. DiffPure (Nie et al., 2022) utilizes diffusion models to purify adversarial perturbations by first adding Gaussian noises to input images and then denoising the images. Diffusion models can also help to improve the certified robustness with randomized smoothing (Carlini et al., 2023; Xiao et al., 2023). Besides, using data generated by diffusion models can significantly improve the performance of adversarial training (Rebuffi et al., 2021; Wang et al., 2023a). However, DiffPure is vulnerable to stronger adaptive attacks while adversarial training models do not generalize well across different threat models, as shown in Table 1. A potential reason of their problems is that they still focus on discriminative classifiers, which do not capture the underlying structure of data distribution. As diffusion models have more accurate score estimation in the whole data space, we aim to explore whether a diffusion model itself can be leveraged to build a robust classifier.

## 3 METHODOLOGY

In this section, we present our **Robust Diffusion Classifier (RDC)**, a robust (generative) classifier constructed from a pre-trained diffusion model. We first provide an overview of diffusion models in Sec. 3.1, then present how to convert a (class-conditional) diffusion model into a classifier in Sec. 3.2 with a robustness analysis considering the optimal setting in Sec. 3.3, and finally detail the likelihood maximization and time complexity reduction techniques to further improve the robustness and efficiency in Sec. 3.4 and Sec. 3.5, respectively. Fig. 1 illustrates our approach.

### 3.1 PRELIMINARY: DIFFUSION MODELS

We briefly review discrete-time diffusion models (Ho et al., 2020). Given a data distribution $q(\mathbf{x})$, the forward diffusion process gradually adds Gaussian noise to the data to obtain a sequence of noisy samples $\{\mathbf{x}_t\}_{t=1}^T$ according to a scaling schedule $\{\alpha_t\}_{t=1}^T$ and a noise schedule $\{\sigma_t\}_{t=1}^T$ as

$$q(\mathbf{x}_t|\mathbf{x}_0) = \mathcal{N}(\mathbf{x}_t; \sqrt{\alpha_t}\mathbf{x}_0, \sigma_t^2\mathbf{I}). \tag{1}$$

Assume that the signal-to-noise ratio $\text{SNR}(t) = \alpha_t/\sigma_t^2$ is strictly monotonically decreasing in time, the sample $\mathbf{x}_t$ is increasingly noisy during the forward process. The scaling and noise schedules are prescribed such that $\mathbf{x}_T$ is nearly an isotropic Gaussian distribution.

To generate images, we need to reverse the diffusion process. It is defined as a Markov chain with learned Gaussian distributions as

$$p_\theta(\mathbf{x}_{0:T}) = p(\mathbf{x}_T)\prod_{t=1}^{T} p_\theta(\mathbf{x}_{t-1}|\mathbf{x}_t), \quad p_\theta(\mathbf{x}_{t-1}|\mathbf{x}_t) = \mathcal{N}(\mathbf{x}_{t-1}; \boldsymbol{\mu}_\theta(\mathbf{x}_t,t), \tilde{\sigma}_t^2\mathbf{I}), \tag{2}$$

where $p(\mathbf{x}_T) = \mathcal{N}(\mathbf{x}_T; \mathbf{0}, \mathbf{I})$ is a prior distribution. Instead of directly parameterizing $\boldsymbol{\mu}_\theta$ via neural networks, it is a common practice (Ho et al., 2020; Kingma et al., 2021) to rewrite $\boldsymbol{\mu}_\theta$ as

$$\boldsymbol{\mu}_\theta(\mathbf{x}_t,t) = \sqrt{\frac{\alpha_{t-1}}{\alpha_t}}\left(\mathbf{x}_t - \sqrt{\frac{\sigma_t}{1-\alpha_t}}\boldsymbol{\epsilon}_\theta(\mathbf{x}_t,t)\right), \tag{3}$$

and learn the time-conditioned noise prediction network $\boldsymbol{\epsilon}_\theta(\mathbf{x}_t,t)$. It can be learned by optimizing the variational lower bound on log-likelihood as

$$\log p_\theta(\mathbf{x}) \geq \mathbb{E}_q[-D_{\text{KL}}(q(\mathbf{x}_T|\mathbf{x}_0)\|p(\mathbf{x}_T)) - \sum_{t>1} D_{\text{KL}}(q(\mathbf{x}_{t-1}|\mathbf{x}_t,\mathbf{x}_0)\|p_\theta(\mathbf{x}_{t-1}|\mathbf{x}_t)) + \log p_\theta(\mathbf{x}_0|\mathbf{x}_1)]$$
$$= -\mathbb{E}_{\boldsymbol{\epsilon},t}\left[w_t\|\boldsymbol{\epsilon}_\theta(\mathbf{x}_t,t) - \boldsymbol{\epsilon}\|_2^2\right] + C_1, \tag{4}$$

where $\mathbb{E}_{\boldsymbol{\epsilon},t}[w_t\|\boldsymbol{\epsilon}_\theta(\mathbf{x}_t,t) - \boldsymbol{\epsilon}\|_2^2]$ is called *diffusion loss* (Kingma et al., 2021), $\boldsymbol{\epsilon}$ follows the standard Gaussian distribution $\mathcal{N}(\mathbf{0}, \mathbf{I})$, $\mathbf{x}_t = \sqrt{\alpha_t}\mathbf{x}_0 + \sigma_t\boldsymbol{\epsilon}$ given by Eq. (1), $C_1$ is typically small and can be dropped (Ho et al., 2020; Song et al., 2021), and $w_t = \frac{\sigma_t\alpha_{t-1}}{2\tilde{\sigma}_t^2(1-\alpha_t)\alpha_t}$. In practice, we set $w_t = 1$ for convenience and good performance as in (Ho et al., 2020).

The conditional diffusion model $p_\theta(\mathbf{x}|y)$ can be parameterized by $\boldsymbol{\epsilon}_\theta(\mathbf{x}_t,t,y)$, while the unconditional model $p_\theta(\mathbf{x})$ can be viewed as a special case with a null input as $\boldsymbol{\epsilon}_\theta(\mathbf{x}_t,t) = \boldsymbol{\epsilon}_\theta(\mathbf{x}_t,t,y=\varnothing)$. A similar lower bound on the conditional log-likelihood is

$$\log p_\theta(\mathbf{x}|y) \geq -\mathbb{E}_{\boldsymbol{\epsilon},t}\left[w_t\|\boldsymbol{\epsilon}_\theta(\mathbf{x}_t,t,y) - \boldsymbol{\epsilon}\|_2^2\right] + C, \tag{5}$$

where $C$ is another small constant that is negligible.

## 3.2 DIFFUSION MODEL FOR CLASSIFICATION

Given an input $\mathbf{x}$, a classifier computes the probability $p_\theta(y|\mathbf{x})$ for all classes $y \in \{1, 2, .., K\}$ with $K$ being the number of classes and outputs the most probable class as $\tilde{y} = \arg\max_y p_\theta(y|\mathbf{x})$. Popular discriminative approaches train Convolutional Neural Networks (Krizhevsky et al., 2012; He et al., 2016) or Vision Transformers (Dosovitskiy et al., 2020; Liu et al., 2021) to directly learn the conditional probability $p_\theta(y|\mathbf{x})$. However, they cannot predict accurately for adversarial example $\mathbf{x}^*$ that is close to the real example $\mathbf{x}$ under the $\ell_p$ norm as $\|\mathbf{x}^* - \mathbf{x}\|_p \leq \epsilon_p$, since it is hard to control how inputs are classified outside the training distribution (Schott et al., 2019).

On the other hand, diffusion models are trained to provide accurate density estimation over the entire data space (Song & Ermon, 2019; Song et al., 2021). By transforming a diffusion model into a generative classifier through Bayes' theorem as $p_\theta(y|\mathbf{x}) \propto p_\theta(\mathbf{x}|y)p(y)$, we hypothesize that the classifier can also give a more accurate conditional probability $p_\theta(y|\mathbf{x})$ in the data space, leading to better adversarial robustness. In this paper, we assume a uniform prior $p(y) = \frac{1}{k}$ for simplicity, which is common for most of the datasets (Krizhevsky & Hinton, 2009; Russakovsky et al., 2015). We show how to compute the conditional probability $p_\theta(y|\mathbf{x})$ via a diffusion model in the following theorem.

**Theorem 3.1.** *(Proof in Appendix A.1) Let $d(\mathbf{x}, y, \theta) = \log p_\theta(\mathbf{x}|y) + \mathbb{E}_{\boldsymbol{\epsilon},t}[w_t\|\boldsymbol{\epsilon}_\theta(\mathbf{x}_t,t,y) - \boldsymbol{\epsilon}\|_2^2]$ denote the gap between the log-likelihood and the diffusion loss. Assume that $y$ is uniformly distributed as $p(y) = \frac{1}{K}$ and $\forall y, d(\mathbf{x}, y, \theta) \to 0$. The conditional probability $p_\theta(y|\mathbf{x})$ can be approximated by*

$$p_\theta(y|\mathbf{x}) = \frac{\exp(-\mathbb{E}_{\boldsymbol{\epsilon},t}[w_t\|\boldsymbol{\epsilon}_\theta(\mathbf{x}_t,t,y) - \boldsymbol{\epsilon}\|_2^2])}{\sum_{\hat{y}}\exp(-\mathbb{E}_{\boldsymbol{\epsilon},t}[w_t\|\boldsymbol{\epsilon}_\theta(\mathbf{x}_t,t,\hat{y}) - \boldsymbol{\epsilon}\|_2^2])}. \tag{6}$$

In Theorem 3.1, we approximate the conditional log-likelihood with the (negative) diffusion loss, which holds true when the gap $d(\mathbf{x}, y, \theta)$ is 0. In practice, although there is inevitably a gap between the log-likelihood and the diffusion loss, we show that the approximation works well in experiments. Eq. (6) requires calculating the noise prediction error over the expectation of random noise $\epsilon$ and timestep $t$, which is efficiently estimated with the variance reduction technique introduced in Sec. 3.5. Besides, our method is also applicable when the prior $p(y)$ is not uniform if we add the logit of class $y$ by $\log p(y)$, where $p(y)$ can be estimated from the training data. Below, we provide an analysis on the adversarial robustness of the diffusion classifier in Eq. (6) under the optimal setting.

## 3.3 ROBUSTNESS ANALYSIS UNDER THE OPTIMAL SETTING

To provide a deeper understanding of the robustness of our diffusion classifier, we first derive the optimal solution of the diffusion model (*i.e.*, diffusion model that has minimal diffusion loss over both the training set and the test set), as shown in the following theorem.

**Theorem 3.2.** *(Proof in Appendix A.2.1) Let $D$ denote a set of examples and $D_y \subset D$ denote a subset of examples whose ground-truth label is $y$. The optimal diffusion model $\epsilon_{\theta_D^*}(\mathbf{x}_t, t, y)$ on the set $D$ is*

$$\epsilon_{\theta_D^*}(\mathbf{x}_t, t, y) = \sum_{\mathbf{x}^{(i)} \in D_y} \frac{1}{\sigma_t}(\mathbf{x}_t - \sqrt{\alpha_t}\mathbf{x}^{(i)}) \cdot \frac{\exp(-\frac{1}{2\sigma_t^2}\|\mathbf{x}_t - \sqrt{\alpha_t}\mathbf{x}^{(i)}\|_2^2)}{\sum_{\mathbf{x}^{(j)} \in D_y} \exp(-\frac{1}{2\sigma_t^2}\|\mathbf{x}_t - \sqrt{\alpha_t}\mathbf{x}^{(j)}\|_2^2)}, \quad (7)$$

Given the optimal diffusion model in Eq. (7), we can easily obtain the optimal diffusion classifier by substituting the solution in Eq. (7) into Eq. (6).

**Corollary 3.3.** *(Proof in Appendix A.2.2) The conditional probability $p_{\theta_D^*}(y|\mathbf{x})$ given the optimal diffusion model $\epsilon_{\theta_D^*}(\mathbf{x}_t, t, y)$ is*

$$p_{\theta_D^*}(y|\mathbf{x}) = \text{softmax}\left(f_{\theta_D^*}(\mathbf{x})\right)_y, \quad f_{\theta_D^*}(\mathbf{x})_y = -\mathbb{E}_{\epsilon,t}\left[\frac{\alpha_t}{\sigma_t^2}\Big\|\sum_{\mathbf{x}^{(i)} \in D_y} s(\mathbf{x}, \mathbf{x}^{(i)}, \epsilon, t)(\mathbf{x} - \mathbf{x}^{(i)})\Big\|_2^2\right],$$

*where*

$$s(\mathbf{x}, \mathbf{x}^{(i)}, \epsilon, t) = \frac{\exp\left(-\frac{1}{2\sigma_t^2}\|\sqrt{\alpha_t}\mathbf{x} + \sigma_t\epsilon - \sqrt{\alpha_t}\mathbf{x}^{(i)}\|_2^2\right)}{\sum_{\mathbf{x}^{(j)} \in D_y} \exp\left(-\frac{1}{2\sigma_t^2}\|\sqrt{\alpha_t}\mathbf{x} + \sigma_t\epsilon - \sqrt{\alpha_t}\mathbf{x}^{(j)}\|_2^2\right)}.$$

**Remark.** Intuitively, the optimal diffusion classifier utilizes the $\ell_2$ norm of the weighted average difference between the input example $\mathbf{x}$ and the real examples $\mathbf{x}^{(i)}$ of class $y$ to calculate the logit for $\mathbf{x}$. The classifier will predict a label $\tilde{y}$ for an input $\mathbf{x}$ if it lies more closely to real examples belonging to $D_{\tilde{y}}$. Moreover, the $\ell_2$ norm is averaged over $t$ with weight $\frac{\alpha_t}{\sigma_t^2}$. As $\frac{\alpha_t}{\sigma_t^2}$ is monotonically decreasing w.r.t. $t$, the classifier gives small weights for noisy examples and large weights for clean examples, which is reasonable since the noisy examples do not play an important role in classification.

Empirically, we evaluate the robust accuracy of the optimal diffusion classifier under the $\ell_\infty$ norm with $\epsilon_\infty = 8/255$ and the $\ell_2$ norm with $\epsilon_2 = 0.5$. We find that the optimal diffusion classifier achieves 100% robust accuracy in both cases, validating our hypothesis that accurate density estimation of diffusion models facilitates robust classification. However, the diffusion models are not optimal in practice. Our trained diffusion classifier can only achieve 35.94% and 76.95% robust accuracy under the $\ell_\infty$ and $\ell_2$ threats, as shown in Table 1. Despite the non-trivial performance without adversarial training, it still lags behind the state-of-the-art. To figure out the problem, we find that the diffusion loss $\mathbb{E}_{\epsilon,t}[w_t\|\epsilon_\theta(\mathbf{x}_t, t, y) - \epsilon\|_2^2]$ is very large for some adversarial example with the ground-truth class. This is caused by either the inaccurate density estimation of $p_\theta(\mathbf{x}|y)$ of the diffusion model or the large gap between the log-likelihood and the diffusion loss violating $d(\mathbf{x}, y, \theta) \to 0$. Developing a better conditional diffusion model can help to address this issue, but we leave this to future work. In the following section, we propose an optimization-based algorithm as an alternative strategy to solve the problem with off-the-shelf diffusion models.

## 3.4 LIKELIHOOD MAXIMIZATION

To address the aforementioned problem, a straightforward approach is to minimize the diffusion loss $\mathbb{E}_{\epsilon,t}[w_t\|\epsilon_\theta(\mathbf{x}_t, t, y) - \epsilon\|_2^2]$ w.r.t. $\mathbf{x}$ such that the input can escape from the region that the diffusion

model cannot provide an accurate density estimation or the gap between the likelihood and diffusion loss $d(\mathbf{x}, y, \theta)$ is large. However, we do not know the ground-truth label of $\mathbf{x}$, making the optimization infeasible. As an alternative strategy, we propose to minimize the unconditional diffusion loss as

$$\min_{\hat{\mathbf{x}}} \mathbb{E}_{\boldsymbol{\epsilon},t}[w_t \|\boldsymbol{\epsilon}_\theta(\hat{\mathbf{x}}_t, t) - \boldsymbol{\epsilon}\|_2^2], \quad \text{s.t. } \|\hat{\mathbf{x}} - \mathbf{x}\|_\infty \leq \eta, \tag{8}$$

where we restrict the $\ell_\infty$ norm between the optimized input $\hat{\mathbf{x}}$ and the original input $\mathbf{x}$ to be smaller than $\eta$, in order to avoid optimizing $\hat{\mathbf{x}}$ into the region of other classes. As Eq. (8) actually maximizes the lower bound of the log-likelihood in Eq. (4), we call this approach **Likelihood Maximization**.

This method can also be viewed as a new diffusion-based purification defense. On one hand, Xiao et al. (2023) prove that for purification defense, a higher likelihood and a smaller distance to the real data of purified input $\hat{\mathbf{x}}$ tends to result in better robustness. Compared to DiffPure, our method restricts the optimization budget by $\eta$, leading to a smaller distance to the real data. Besides, unlike DiffPure which only maximizes the likelihood with a high probability (Xiao et al., 2023), we directly maximize the likelihood, leading to improved robustness. On the other hand, because the adversarial example usually lies in the vicinity of its corresponding real example of the ground-truth class $y$, moving along the direction towards higher $\log p(\mathbf{x})$ will probably lead to higher $\log p(\mathbf{x}|y)$. Therefore, the optimized input $\hat{\mathbf{x}}$ could be more accurately classified by the diffusion classifier.

### 3.5 TIME COMPLEXITY REDUCTION

**Accelerating diffusion classifier.** A common practice for estimating the diffusion loss in Eq. (6) is to adopt the Monte Carlo sampling. However, this will lead to a high variance with few samples or high time complexity with many samples. To reduce the variance with affordable computational cost, we directly compute the expectation over $t$ instead of sampling $t$ as

$$\mathbb{E}_{\boldsymbol{\epsilon},t}[w_t \|\boldsymbol{\epsilon}_\theta(\mathbf{x}_t, t, y) - \boldsymbol{\epsilon}\|_2^2] = \frac{1}{T} \sum_{t=1}^{T} \mathbb{E}_{\boldsymbol{\epsilon}}[w_t \|\boldsymbol{\epsilon}_\theta(\mathbf{x}_t, t, y) - \boldsymbol{\epsilon}\|_2^2]. \tag{9}$$

Eq. (9) requires to calculate the noise prediction error for all timesteps. For $\boldsymbol{\epsilon}$, we still adopt Monte Carlo sampling, but we show that sampling only one $\boldsymbol{\epsilon}$ is sufficient to achieve good performance. We can further reduce the number of timesteps by systematic sampling that selects the timesteps at a uniform interval. Although it does not lead to an obvious drop in clean accuracy, it will significantly affect robust accuracy as shown in Sec. 4.5, because the objective is no longer strongly correlated with log-likelihood after reducing the number of timesteps.

Our diffusion classifier in Eq. (6) requires $K \times T$ NFEs (Number of Function Evaluations), which limits the applicability to large datasets. This is because current diffusion models are mainly designed for image generation tasks, they can only provide predictions for one class at a time. To obtain the predictions of all classes in a single forward pass, we propose to modify the last convolutional layer in the UNet backbone to output predictions for $K$ classes (i.e., $K \times 3$ dimensions) simultaneously, which only requires $T$ NFEs for a single image. We name this set of novel diffusion backbones as **multi-head diffusions**. For more details, please refer to Appendix B.1.

**Accelerating likelihood maximization.** To further reduce the time complexity of likelihood maximization, for each iteration, instead of calculating the diffusion loss using all timesteps like Eq. (9), we only uniformly sample a single timestep to approximate the expectation of the diffusion loss. Surprisingly, this modification not only reduces the time complexity of likelihood maximization from $O(N \times T)$ to $O(N)$, but also greatly boosts the robustness. This is because this likelihood maximization induces more randomness, thus it is more effective to smooth the local extrema. We provide more in-depth analysis in Appendix B.2.

Given the above techniques, the overall algorithm of RDC is outlined in Algorithm 1.

## 4 EXPERIMENTS

In this section, we first provide the experimental settings in Sec. 4.1. We then show the effectiveness of our method compared with the state-of-the-art methods in Sec. 4.2 and the generalizability across different threat models in Sec. 4.3. We provide thorough analysis to examine gradient obfuscation in Sec. 4.4 and conduct various ablation studies in Sec. 4.5.

---

**Algorithm 1** Robust Diffusion Classifier (RDC)

---

**Require:** A pre-trained diffusion model $\epsilon_\theta$, input image $\mathbf{x}$, optimization budget $\eta$, step size $\gamma$, optimization steps $N$, momentum decay factor $\mu$.

1: **Initialize:** $\mathbf{m} = 0, \hat{\mathbf{x}} = \mathbf{x}$;
2: **for** $n = 0$ to $N - 1$ **do**
3:     Estimate $\mathbf{g} = \nabla_{\mathbf{x}} \mathbb{E}_{\epsilon,t}[w_t \|\epsilon_\theta(\hat{\mathbf{x}}_t, t) - \epsilon\|_2^2]$ using one randomly sampled $t$ and $\epsilon$;
4:     Update momentum $\mathbf{m} = \mu \cdot \mathbf{m} - \frac{\mathbf{g}}{\|\mathbf{g}\|_1}$;
5:     Update $\hat{\mathbf{x}}$ by $\hat{\mathbf{x}} = \text{clip}_{\mathbf{x},\eta}(\hat{\mathbf{x}} + \gamma \cdot \mathbf{m})$;
6: **end for**
7: Calculate $\mathbb{E}_{\epsilon,t}[w_t \|\epsilon_\theta(\hat{\mathbf{x}}_t, t, y) - \epsilon\|_2^2]$ for all $y \in \{1, 2, ..., K\}$ simultaneously using multi-head diffusion;
8: Calculate $p_\theta(y|\mathbf{x})$ by Eq. (6);
9: **Return:** $\tilde{y} = \arg\max_y p_\theta(y|\mathbf{x})$.

---

Table 1: Clean accuracy (%) and robust accuracy (%) of different methods against unseen threats.

| Method | Architecture | Clean Acc | Robust Acc | | | |
| --- | --- | --- | --- | --- | --- | --- |
| | | | $\ell_\infty$ norm | $\ell_2$ norm | StAdv | Avg |
| AT-DDPM-$\ell_\infty$ | WRN70-16 | 88.87 | 63.28 | 64.65 | 4.88 | 44.27 |
| AT-DDPM-$\ell_2$ | WRN70-16 | 93.16 | 49.41 | 81.05 | 5.27 | 45.24 |
| AT-EDM-$\ell_\infty$ | WRN70-16 | 93.36 | 70.90 | 69.73 | 2.93 | 47.85 |
| AT-EDM-$\ell_2$ | WRN70-16 | 95.90 | 53.32 | **84.77** | 5.08 | 47.72 |
| PAT-self | AlexNet | 75.59 | 47.07 | 64.06 | 39.65 | 50.26 |
| DiffPure ($t^* = 0.125$) | UNet+WRN70-16 | 87.50 | 53.12 | 75.59 | 12.89 | 47.20 |
| DiffPure ($t^* = 0.1$) | UNet+WRN70-16 | 90.97 | 53.52 | 72.65 | 12.89 | 46.35 |
| SBGC | UNet | 95.04 | 0.00 | 0.00 | 0.00 | 0.00 |
| HybViT | ViT | 95.90 | 0.00 | 0.00 | 0.00 | 0.00 |
| JEM | WRN28-10 | 92.90 | 8.20 | 26.37 | 0.05 | 11.54 |
| LM (ours) | UNet+WRN70-16 | 87.89 | 71.68 | 75.00 | 87.50 | **78.06** |
| DC (ours) | UNet | 93.55 | 35.94 | 76.95 | **93.55** | 68.81 |
| RDC (LM+DC) (ours) | UNet | 89.85 | **75.67** | 82.03 | 89.45 | **82.38** |

## 4.1 EXPERIMENTAL SETTINGS

**Datasets and training details.** Following Nie et al. (2022), we randomly select 512 images from the CIFAR-10 test set (Krizhevsky & Hinton, 2009) for evaluation due to the high computational cost of the attack algorithms. We also conduct experiments on other datasets and other settings in Appendix B.2. We adopt off-the-shelf conditional diffusion model in Karras et al. (2022) and train our multi-head diffusion as detailed in Appendix B for 100 epochs on CIFAR-10 training set.

**Hyperparameters.** In likelihood maximization, we set the optimization steps $N = 5$, momentum decay factor $\mu = 1$, optimization budget $\eta = 8/255$ (see Sec. 4.5 for an ablation study), step size $\gamma = 0.1$. For each timestep, we only sample one $\epsilon$ to estimate $\mathbb{E}_\epsilon[w_t \|\epsilon_\theta(\mathbf{x}_t, t, y) - \epsilon\|_2^2]$.

**Robustness evaluation.** Following Nie et al. (2022), we evaluate the clean accuracy and robust accuracy using AutoAttack (Croce & Hein, 2020) under both $\ell_\infty$ norm of $\epsilon_\infty = 8/255$ and $\ell_2$ norm of $\epsilon_2 = 0.5$. To demonstrate the generalization ability towards unseen threat models, we also evaluate the robustness against StAdv (Xiao et al., 2018) with 100 steps under the bound of 0.05. Since computing the gradient through likelihood maximization requires calculating the second-order derivative, we use BPDA (Athalye et al., 2018) as the default adaptive attack, approximating the gradient with an identity mapping. We conduct more comprehensive evaluations of gradient obfuscation in Sec. 4.4, where we show that BPDA is as strong as computing the exact gradient.

## 4.2 COMPARISON WITH THE STATE-OF-THE-ART

We compare our method with the state-of-the-art defense methods, including adversarial training with DDPM generated data (AT-DDPM) (Rebuffi et al., 2021), adversarial training with EDM generated data (AT-EDM) (Wang et al., 2023a), and DiffPure (Nie et al., 2022). We also compare with perceptual adversarial training (PAT-self) (Laidlaw et al., 2021) and other generative classifiers, including SBGC (Zimmermann et al., 2021), HybViT (Yang et al., 2022), and JEM (Grathwohl et al., 2019). Notably,

robust accuracy of most baselines does not change much on our selected subset. We also compare the time complexity and robustness of our model with more methods in Table 3 in Appendix B.2.

DiffPure incurs significant memory usage and substantial randomness, posing challenges for robustness evaluation. Their proposed adjoint method (Nie et al., 2022) is insufficient to measure the model robustness. To mitigate this issue, we employ gradient checkpoints to compute the exact gradient and leverage Expectation Over Time (EOT) to reduce the impact of randomness during optimization. Rather than using the 640 times EOT recommended in Fig. 2(a), we adopt PGD-80 (Madry et al., 2018) with 10 times EOT and a large step size $1/255$ to efficiently evaluate DiffPure.

Table 1 shows the results of Likelihood Maximization (LM), Diffusion Classifier (DC) and Robust Diffusion Classifier (RDC) compared with baselines under the $\ell_\infty$ and $\ell_2$ norm threat models. We can see that the robustness of DC outperforms all previous generative classifiers by a large margin. Specifically, DC improves the robust accuracy over JEM by +27.74% under the $\ell_\infty$ norm and +50.58% under the $\ell_2$ norm. RDC can further improve the performance over DC, which achieves 75.67% and 82.03% robust accuracy under the two settings. Notably, RDC outperforms the previous state-of-the-art model AT-EDM (Wang et al., 2023a) by +4.77% under the $\ell_\infty$ norm.

### 4.3 Defense against unseen threats

Adversarial training methods often suffer from poor generalization across different threat models, while DiffPure requires adjusting purification noise scales for different threat models, which limits their applicability in real-world scenarios where the threat models are unknown. In contrast, our proposed methods are agnostic to specific threat models. To demonstrates the strong generalization ability of our methods across different threat models, we evaluate the generalization performance of our proposed method by testing against different threats, including $\ell_\infty$, $\ell_2$, and StAdv.

Table 1 presents the results, demonstrating that the average robustness of our methods surpass the baselines by more than 30%. Specifically, RDC outperforms $\ell_\infty$ adversarial training models by +12.30% under the $\ell_2$ norm and $\ell_2$ adversarial training models by +22.35% under the $\ell_\infty$ norm. Impressively, LM, DC and RDC achieve 87.50%, 93.55% and 89.45% robustness under StAdv, surpassing previous methods by more than 53.90%. These results indicate the strong generalization ability of our method and its potential to be applied in real-world scenarios under unknown threats.

### 4.4 Evaluation of gradient obfuscation

Diffusion Classifier can be directly evaluated by AutoAttack. However, Robust Diffusion Classifier could not be directly evaluated by AutoAttack due to the indifferentiable likelihood maximization. To demonstrate the effectiveness of BPDA adaptive attack, we conduct the following experiments.

**Exact gradient attack.** To directly evaluate the robustness of RDC, we utilize gradient checkpoint and create a computational graph during backpropagation to obtain exact gradients. However, we could only evaluate RDC when $N = 1$ due to the large memory cost. As shown in Table 2, our RDC with $N = 1$ achieves 69.53% robust accuracy under the exact gradient attack, about 0.39% lower than BPDA. This result suggests that BPDA suffices for evaluating RDC.

Table 2: Robust accuracy (%) of RDC under different adaptive attacks. BPDA ($N = 5$) and Lagrange ($N = 5$) are adaptive attacks for RDC with 5 steps of LM, while Exact Gradient ($N = 1$) and BPDA ($N = 1$) are adaptive attacks for RDC with 1 step.

| Attack | Robust Acc |
|---|---|
| BPDA ($N = 5$) | 75.67 |
| Lagrange ($N = 5$) | 77.54 |
| Exact Gradient ($N = 1$) | 69.53 |
| BPDA ($N = 1$) | 69.92 |

**Lagrange attack.** RDC optimizes the unconditional diffusion loss before feeding the inputs into DC. If our adversarial examples already have a small unconditional diffusion loss or a large $\log p(\mathbf{x})$, it may not be interrupted during likelihood maximization. Therefore, to produce adversarial examples with a small diffusion loss, we set our loss function as

$$\text{Lagrange}(\mathbf{x}, y, \boldsymbol{\epsilon}_\theta) = \log p_\theta(y|\mathbf{x}) + l \cdot \mathbb{E}_{\boldsymbol{\epsilon}, t}[w_t \|\boldsymbol{\epsilon}_\theta(\mathbf{x}_t, t) - \boldsymbol{\epsilon}\|_2^2], \tag{10}$$

where $p_\theta(y|\mathbf{x})$ is given by Eq. (6), and the first term is the (negative) cross-entropy loss to induce misclassification. For an input, we craft adversarial examples using three different weights, $l = 0, 1, 10$. If one of these three loss functions successfully generate an adversarial example, we count it as a successful attack. As shown in Table 2, this adaptive attack is no more effective than BPDA.

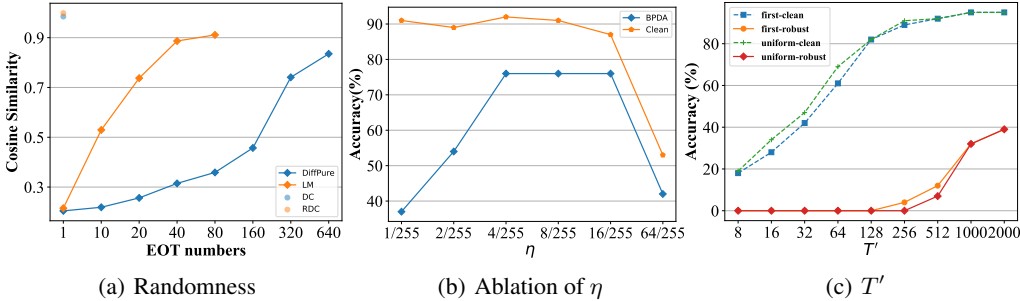

|(a) Randomness|(b) Ablation of $\eta$|(c) $T'$|

Figure 2: (a-b): Impact of $T'$ and $\eta$ in our method on standard accuracy and robust accuracy against our BPDA adaptive attack. (c): Randomness of different methods.

**Gradient randomness.** To quantify the randomness, we compute the gradients of each model w.r.t. the input ten times and compute the pairwise cosine similarity between the gradients. We then average these cosine similarities across 100 images. To capture the randomness when using EOT, we treat the gradients obtained after applying EOT as a single time and repeat the same process to compute their cosine similarity. As shown in Fig. 2(a), the gradients of our methods exhibit low randomness, while DiffPure is more than 640 times as random as DC, RDC, and about 16 times as random as LM. Thus, the robustness of our methods is not primarily due to the stochasticity of gradients.

## 4.5 Ablation studies

In this section, we perform ablation studies of several hyperparameters in our algorithm with the first 100 examples in the CIFAR-10 test set. All the experiments are done under AutoAttack with BPDA of $\ell_\infty$ bounded perturbations with $\epsilon_\infty = 8/255$.

**Optimization budget $\eta$.** To find the best optimization budget $\eta$, we test the robust accuracy of different optimization budgets. As shown in Fig. 2(b), the robust accuracy first increases and then decreases as $\eta$ becomes larger. When $\eta$ is small, we could not move $\mathbf{x}$ out of the adversarial region. However, when $\eta$ is too large, we may optimize $\mathbf{x}$ into an image of another class. Therefore, we should choose an appropriate $\eta$. In this work, we set $\eta = 8/255$.

**Sampling timesteps.** We also attempt to reduce the number of timesteps used in calculating the diffusion loss. Since only the DC is influenced by this parameter, we conduct this experiment exclusively on DC to minimize the impact of other factors. One way is to only calculate the diffusion loss of the first $T'$ timesteps $\{i\}_{i=1}^{T'}$ ("first-clean" and "first-robust" in Fig. 2(c)). Inspired by Song et al. (2020), another way is to use systematic sampling, where we use timesteps $\{iT/T'\}_{i=1}^{T'}$ ("uniform-clean" and "uniform-robust" in Fig. 2(c) ). Both methods achieve similar results on clean accuracy and robust accuracy. Although a significant reduction of $T'$ does not lead to an obvious drop in clean accuracy, it will significantly affect robust accuracy due to the reason discussed in Sec. 3.5.

**Sampling steps for $\epsilon$.** We also attempt to improve the estimation of $\mathbb{E}_\epsilon[w_t\|\epsilon_\theta(\mathbf{x}_t, t, y) - \epsilon\|_2^2]$ by sampling $\epsilon$ multiple times or keeping $\epsilon$ the same for different timesteps or different classes. However, these increase neither robustness nor accuracy because we have already computed $T$ times for the expectation over $t$. From another perspective, the cosine similarity of the gradients is about 98.48%, suggesting that additional sampling of $\epsilon$ or using the same $\epsilon$ is unnecessary.

## 5 Conclusion

In this paper, we propose a novel defense method called Robust Diffusion Classifier (RDC), which leverages a single diffusion model to directly classify input images by predicting data likelihood by diffusion model and calculating class probabilities through Bayes' theorem. We theoretically analyze the robustness of our diffusion classifier, propose to maximize the log-likelihood before feeding the input images into the diffusion classifier. We also propose multi-head diffusion which greatly reduces the time complexity of RDC. To demonstrate the performance of our method, we evaluate our method with strong adaptive attacks and conduct extensive experiments. Our method achieves state-of-the-art robustness against these strong adaptive attacks and generalizes well to unseen threat models.

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

## A  PROOFS AND DERIVATIONS

### A.1  PROOF OF THEOREM 3.1

*Proof.*

$$
\begin{aligned}
p_\theta(y|\mathbf{x}) &= \frac{p_\theta(\mathbf{x}, y)}{\sum_{\hat{y}} p_\theta(\mathbf{x}, \hat{y})} \\
&= \frac{p_\theta(\mathbf{x}|y)p_\theta(y)}{\sum_y p_\theta(\mathbf{x}|\hat{y})p_\theta(\hat{y})} \\
&= \frac{p_\theta(\mathbf{x}|y)}{\sum_{\hat{y}} p_\theta(\mathbf{x}|\hat{y})} \\
&= \frac{e^{\log p_\theta(\mathbf{x}|y)}}{\sum_{\hat{y}} e^{\log p_\theta(\mathbf{x}|\hat{y})}} \\
&= \frac{\exp\left(\mathbb{E}_{\boldsymbol{\epsilon},t}[w_t\|\boldsymbol{\epsilon}_\theta(\mathbf{x}_t, t, y) - \boldsymbol{\epsilon}\|_2^2] + \log p_\theta(\mathbf{x}|y) - \mathbb{E}_{\boldsymbol{\epsilon},t}[w_t\|\boldsymbol{\epsilon}_\theta(\mathbf{x}_t, t, y) - \boldsymbol{\epsilon}\|_2^2]\right)}{\sum_{\hat{y}} \exp\left(\mathbb{E}_{\boldsymbol{\epsilon},t}[w_t\|\boldsymbol{\epsilon}_\theta(\mathbf{x}_t, t, \hat{y}) - \boldsymbol{\epsilon}\|_2^2] + \log p_\theta(\mathbf{x}|\hat{y}) - \mathbb{E}_{\boldsymbol{\epsilon},t}[w_t\|\boldsymbol{\epsilon}_\theta(\mathbf{x}_t, t, \hat{y}) - \boldsymbol{\epsilon}\|_2^2]\right)} \\
&= \frac{\exp\left(d(\mathbf{x}, y, \theta)\right)\exp\left(-\mathbb{E}_{\boldsymbol{\epsilon},t}[w\|\boldsymbol{\epsilon}_\theta(\mathbf{x}_t, t, y) - \boldsymbol{\epsilon}\|_2^2]\right)}{\sum_{\hat{y}} \exp\left(d(\mathbf{x}, \hat{y}, \theta)\right)\exp\left(-\mathbb{E}_{\boldsymbol{\epsilon},t}[w_t\|\boldsymbol{\epsilon}_\theta(\mathbf{x}_t, t, \hat{y}) - \boldsymbol{\epsilon}\|_2^2]\right)}.
\end{aligned}
$$

When $\forall \hat{y}, \ d(\mathbf{x}, \hat{y}, \theta) \to 0$, we can get:

$$
\forall \hat{y}, \ \exp\left(\mathbb{E}_{\boldsymbol{\epsilon},t}[w_t\|\boldsymbol{\epsilon}_\theta(\mathbf{x}_t, t, \hat{y}) - \boldsymbol{\epsilon}\|_2^2] + \log p_\theta(\mathbf{x}|\hat{y})\right) \to 1.
$$

Therefore,

$$
p_\theta(y|\mathbf{x}) = \frac{\exp\left(-\mathbb{E}_{\boldsymbol{\epsilon},t}[w_t\|\boldsymbol{\epsilon}_\theta(\mathbf{x}_t, t, y) - \boldsymbol{\epsilon}\|_2^2]\right)}{\sum_{\hat{y}} \exp\left(-\mathbb{E}_{\boldsymbol{\epsilon},t}[w_t\|\boldsymbol{\epsilon}_\theta(\mathbf{x}_t, t, \hat{y}) - \boldsymbol{\epsilon}\|_2^2]\right)}.
$$

$\square$

### A.2  DERIVATION OF THE OPTIMAL DIFFUSION CLASSIFIER

#### A.2.1  OPTIMAL DIFFUSION MODEL: PROOF OF THEOREM 3.2

*Proof.* The optimal diffusion model has the minimal error $\mathbb{E}_{\mathbf{x},t,y}[\|\boldsymbol{\epsilon}(\mathbf{x}_t, t, y) - \boldsymbol{\epsilon}\|_2^2]$ among all the models in hypothesis space. Since the prediction for one input pair $(\mathbf{x}_t, t, y)$ does not affect the prediction for any other input pairs, the optimal diffusion model will give the optimal solution for any input pair $(\mathbf{x}_t, t, y)$:

$$
\mathbb{E}_{\mathbf{x}^{(i)} \sim p(\mathbf{x}^{(i)}|\mathbf{x}_t, y)}[\|\boldsymbol{\epsilon}_{\theta_D^*}(\mathbf{x}_t, t, y) - \boldsymbol{\epsilon}_i\|_2^2] = \min_\theta \mathbb{E}_{\mathbf{x}^{(i)} \sim p(\mathbf{x}^{(i)}|\mathbf{x}_t, y)}[\|\boldsymbol{\epsilon}_\theta(\mathbf{x}_t, t, y) - \boldsymbol{\epsilon}_i\|_2^2],
$$

where $\boldsymbol{\epsilon}_i = \frac{\mathbf{x}_t - \sqrt{\alpha_t}\mathbf{x}^{(i)}}{\sigma_t}$.

Note that

$$
p(\mathbf{x}^{(i)}|\mathbf{x}_t, y) = \frac{p(\mathbf{x}^{(i)}|y)p(\mathbf{x}_t|\mathbf{x}^{(i)}, y)}{p(\mathbf{x}_t|y)} = \frac{p(\mathbf{x}^{(i)}|y)q(\mathbf{x}_t|\mathbf{x}^{(i)})}{p(\mathbf{x}_t|y)}.
$$

Assume that

$$
p(\mathbf{x}^{(i)}|y) = \left\{ \begin{array}{ll} \frac{1}{|D_y|} & , \mathbf{x}^{(i)} \in D_y \\ 0 & , \mathbf{x}^{(i)} \notin D_y \end{array} \right.
$$

Solving $\frac{\partial}{\partial \boldsymbol{\epsilon}_\theta(\mathbf{x}_t, t, y)} \mathbb{E}_{\mathbf{x}^{(i)} \sim p(\mathbf{x}^{(i)}|\mathbf{x}_t, y)}[\|\boldsymbol{\epsilon}_\theta(\mathbf{x}_t, t, y) - \boldsymbol{\epsilon}_i\|_2^2] = 0$, we can get:

$$\mathbb{E}_{\mathbf{x}^{(i)} \sim p(\mathbf{x}^{(i)}|\mathbf{x}_t, y)}[\boldsymbol{\epsilon}_\theta(\mathbf{x}_t, t, y) - \boldsymbol{\epsilon}_i] = 0,$$

$$\sum_{x^{(i)} \in D} p(\mathbf{x}^{(i)}|\mathbf{x}_t, y)\boldsymbol{\epsilon}_\theta(\mathbf{x}_t, t, y) = \boldsymbol{\epsilon}_\theta(\mathbf{x}_t, t, y) = \sum_{\mathbf{x}^{(i)} \in D_y} p(\mathbf{x}^{(i)}|\mathbf{x}_t, y)\boldsymbol{\epsilon}_i.$$

Substitute $\boldsymbol{\epsilon}_i$ by Eq. (1):

$$\boldsymbol{\epsilon}_\theta(\mathbf{x}_t, t, y) = \sum_{\mathbf{x}^{(i)} \in D_y} p(\mathbf{x}^{(i)}|\mathbf{x}_t, y)\frac{\mathbf{x}_t - \sqrt{\alpha_t}\mathbf{x}^{(i)}}{\sigma_t}$$

$$= \sum_{\mathbf{x}^{(i)} \in D_y} \frac{p(\mathbf{x}^{(i)}|y)q(\mathbf{x}_t|\mathbf{x}^{(i)})}{p(\mathbf{x}_t|y)}\frac{\mathbf{x}_t - \sqrt{\alpha_t}\mathbf{x}^{(i)}}{\sigma_t}$$

$$= \sum_{\mathbf{x}^{(i)} \in D_y} \frac{p(\mathbf{x}^{(i)}|y)}{p(\mathbf{x}_t|y)}p(\mathcal{N}(\mathbf{x}_t|\sqrt{\alpha_t}\mathbf{x}^{(i)}, \sigma_t^2 I) = \frac{\mathbf{x}_t - \sqrt{\alpha_t}\mathbf{x}^{(i)}}{\sigma_t})\frac{\mathbf{x}_t - \sqrt{\alpha_t}\mathbf{x}^{(i)}}{\sigma_t}$$

$$= \sum_{\mathbf{x}^{(i)} \in D_y} \frac{p(\mathbf{x}^{(i)}|y)}{p(\mathbf{x}_t|y)}\frac{1}{(2\pi\sigma_t)^{\frac{n}{2}}}\exp\left(-\frac{\|\mathbf{x}_t - \sqrt{\alpha_t}\mathbf{x}^{(i)}\|_2^2}{2\sigma_t^2}\right)\frac{\mathbf{x}_t - \sqrt{\alpha_t}\mathbf{x}^{(i)}}{\sigma_t}$$

To avoid numerical problem caused by $\frac{1}{(2\pi\sigma_t)^{\frac{n}{2}}}$ and intractable $\frac{p(\mathbf{x}^{(i)}|y)}{p(\mathbf{x}_t|y)}$, we re-organize this equation using softmax function:

$$\boldsymbol{\epsilon}_\theta(\mathbf{x}_t, t, y) = \sum_{\mathbf{x}^{(i)} \in D_y} \frac{\frac{p(\mathbf{x}^{(i)}|y)}{p(\mathbf{x}_t|y)}\frac{1}{(2\pi\sigma_t)^{\frac{n}{2}}}\exp\left(-\frac{\|\mathbf{x}_t - \sqrt{\alpha_t}\mathbf{x}^{(i)}\|_2^2}{2\sigma_t^2}\right)}{\sum_{j=1}^{|D_y|} p(\mathbf{x}_j|\mathbf{x}_t, y)}\frac{\mathbf{x}_t - \sqrt{\alpha_t}\mathbf{x}^{(i)}}{\sigma_t}$$

$$= \sum_{\mathbf{x}^{(i)} \in D_y} \frac{\frac{p(\mathbf{x}^{(i)}|y)}{p(\mathbf{x}_t|y)}\frac{1}{(2\pi\sigma_t)^{\frac{n}{2}}}\exp\left(-\frac{\|\mathbf{x}_t - \sqrt{\alpha_t}\mathbf{x}^{(i)}\|_2^2}{2\sigma_t^2}\right)}{\sum_{j=1}^{|D_y|} \frac{p(\mathbf{x}_j|y)}{p(\mathbf{x}_t|y)}\frac{1}{(2\pi\sigma_t)^{\frac{n}{2}}}\exp\left(-\frac{\|\mathbf{x}_t - \sqrt{\alpha_t}\mathbf{x}_j\|_2^2}{2\sigma_t^2}\right)}\frac{\mathbf{x}_t - \sqrt{\alpha_t}\mathbf{x}^{(i)}}{\sigma_t}$$

$$= \sum_{\mathbf{x}^{(i)} \in D_y} \frac{1}{\sigma_t}(\mathbf{x}_t - \sqrt{\alpha_t}\mathbf{x}^{(i)})\frac{\exp(-\frac{1}{2\sigma_t^2}\|\mathbf{x}_t - \sqrt{\alpha_t}\mathbf{x}^{(i)}\|_2^2)}{\sum_{\mathbf{x}^{(j)} \in D_y}\exp(-\frac{1}{2\sigma_t^2}\|\mathbf{x}_t - \sqrt{\alpha_t}\mathbf{x}^{(j)}\|_2^2)}.$$

This is the result of Eq. (7). □

### A.2.2 OPTIMAL DIFFUSION CLASSIFIER: PROOF OF THEOREM 3.3

*Proof.* Substitute Eq. (7) into Eq. (6):

$$f_{\theta_D^*}(\mathbf{x})_y$$

$$= -\mathbb{E}_{t,\boldsymbol{\epsilon}}[\|\boldsymbol{\epsilon}_\theta(\mathbf{x}_t, t, y) - \boldsymbol{\epsilon}\|_2^2]$$

$$= -\mathbb{E}_{t,\boldsymbol{\epsilon}}[\|\sum_{\mathbf{x}^{(i)} \in D_y}[\frac{(\mathbf{x}_t - \sqrt{\alpha_t}\mathbf{x}^{(i)})}{\sigma_t}\frac{\exp(-\frac{\|\mathbf{x}_t - \sqrt{\alpha_t}\mathbf{x}^{(i)}\|_2^2}{2\sigma_t^2})}{\sum_{\mathbf{x}^{(j)} \in D_y}\exp(-\frac{\|\mathbf{x}_t - \sqrt{\alpha_t}\mathbf{x}^{(j)}\|_2^2}{2\sigma_t^2})}] - \boldsymbol{\epsilon}\|_2^2]$$

$$= -\mathbb{E}_{t,\boldsymbol{\epsilon}}[\|\sum_{\mathbf{x}^{(i)} \in D_y}[\frac{(\sqrt{\alpha_t}\mathbf{x} + \sigma_t\boldsymbol{\epsilon} - \sqrt{\alpha_t}\mathbf{x}^{(i)})}{\sigma_t}\frac{\exp\left(-\frac{\|\sqrt{\alpha_t}\mathbf{x} + \sigma_t\boldsymbol{\epsilon} - \sqrt{\alpha_t}\mathbf{x}^{(i)}\|_2^2}{2\sigma_t^2}\right)}{\sum_{\mathbf{x}^{(j)} \in D_y}\exp\left(-\frac{\|\sqrt{\alpha_t}\mathbf{x} + \sigma_t\boldsymbol{\epsilon} - \sqrt{\alpha_t}\mathbf{x}^{(j)}\|_2^2}{2\sigma_t^2}\right)}] - \boldsymbol{\epsilon}\|_2^2]$$

$$= -\mathbb{E}_{t,\boldsymbol{\epsilon}}[\|\sum_{\mathbf{x}^{(i)} \in D_y}[\frac{1}{\sigma_t}(\sqrt{\alpha_t}\mathbf{x} - \sqrt{\alpha_t}\mathbf{x}^{(i)})s(\mathbf{x}, \mathbf{x}^{(i)}, \boldsymbol{\epsilon}, t) + \boldsymbol{\epsilon}\sum_{\mathbf{x}^{(i)} \in D_y} s(\mathbf{x}, \mathbf{x}^{(i)}, \boldsymbol{\epsilon}, t)]$$

$$= -\mathbb{E}_{t,\boldsymbol{\epsilon}}[\|\sum_{\mathbf{x}^{(i)} \in D_y}[\frac{1}{\sigma_t}(\sqrt{\alpha_t}\mathbf{x} - \sqrt{\alpha_t}\mathbf{x}^{(i)})s(\mathbf{x}, \mathbf{x}^{(i)}, \boldsymbol{\epsilon}, t)]\|_2^2]$$

$$= -\mathbb{E}_{\boldsymbol{\epsilon},t}[\frac{\alpha_t}{\sigma_t^2}\|\sum_{\mathbf{x}^{(i)} \in D_y} s(\mathbf{x}, \mathbf{x}^{(i)}, \boldsymbol{\epsilon}, t)(\mathbf{x} - \mathbf{x}^{(i)})\|_2^2].$$

We get the result. □

### A.3 DERIVATION OF CONDITIONAL ELBOS IN EQ. (5)

We provide a derivation of conditional ELBO in the following, which is similar to the unconditional ELBO in Ho et al. (2020).

$$
\begin{aligned}
&\log p_\theta(\mathbf{x}_0|y)\\
&= \log \int \frac{p_\theta(\mathbf{x}_{t:T}|y)q(\mathbf{x}_{1:T}|\mathbf{x}_0,y)}{q(\mathbf{x}_{1:T}|\mathbf{x}_0,y)}d\mathbf{x}_{1:T}\\
&= \log \mathbb{E}_{q(\mathbf{x}_{1:T}|\mathbf{x}_0,y)}[\frac{p_\theta(\mathbf{x}_T|y)p_\theta(\mathbf{x}_{t:T-1}|\mathbf{x}_T,y)}{q(\mathbf{x}_{1:T}|\mathbf{x}_0,y)}]\\
&\geq \mathbb{E}_{q(\mathbf{x}_{1:T}|\mathbf{x}_0,y)}[\log \frac{p_\theta(\mathbf{x}_T|y)p_\theta(\mathbf{x}_{t:T-1}|\mathbf{x}_T,y)}{q(\mathbf{x}_{1:T}|\mathbf{x}_0,y)}]\\
&= \mathbb{E}_{q(\mathbf{x}_{1:T}|\mathbf{x}_0,y)}[\log \frac{p_\theta(\mathbf{x}_T|y)\prod_{i=0}^{T-1}p_\theta(\mathbf{x}_i|\mathbf{x}_{i+1},y)}{\prod_{i=0}^{T-1}q(\mathbf{x}_{i+1}|\mathbf{x}_i,\mathbf{x}_0,y)}]\\
&= \mathbb{E}_{q(\mathbf{x}_{1:T}|\mathbf{x}_0,y)}[\log \frac{p_\theta(\mathbf{x}_T|y)\prod_{i=0}^{T-1}p_\theta(\mathbf{x}_i|\mathbf{x}_{i+1},y)}{\prod_{i=0}^{T-1}\frac{q(\mathbf{x}_{i+1}|\mathbf{x}_0,y)q(\mathbf{x}_i|\mathbf{x}_{i+1},\mathbf{x}_0,y)}{q(\mathbf{x}_i|\mathbf{x}_0,y)}}]\\
&= \mathbb{E}_{q(\mathbf{x}_{1:T}|\mathbf{x}_0,y)}[\log \frac{p_\theta(\mathbf{x}_T|y)\prod_{i=0}^{T-1}p_\theta(\mathbf{x}_i|\mathbf{x}_{i+1},y)}{\prod_{i=0}^{T-1}q(\mathbf{x}_i|\mathbf{x}_{i+1},\mathbf{x}_0,y)} - \log q(\mathbf{x}_T|\mathbf{x}_0,y)]\\
&= \mathbb{E}_{q(\mathbf{x}_{1:T}|\mathbf{x}_0,y)}[\log \frac{\prod_{i=0}^{T-1}p_\theta(\mathbf{x}_i|\mathbf{x}_{i+1},y)}{\prod_{i=0}^{T-1}q(\mathbf{x}_i|\mathbf{x}_{i+1},\mathbf{x}_0,y)} - \log \frac{q(\mathbf{x}_T|\mathbf{x}_0,y)}{p_\theta(\mathbf{x}_T|y)}]\\
&= \sum_{i=0}^{T-1}\mathbb{E}_{q(\mathbf{x}_i,\mathbf{x}_{i+1}|\mathbf{x}_0,y)}[\log \frac{p_\theta(\mathbf{x}_i|\mathbf{x}_{i+1},y)}{q(\mathbf{x}_i|\mathbf{x}_{i+1},\mathbf{x}_0,y)}] - D_{KL}(q(\mathbf{x}_T|\mathbf{x}_0,y)\|p_\theta(\mathbf{x}_T|y))\\
&= \sum_{i=0}^{T-1}\mathbb{E}_{q(\mathbf{x}_{i+1}|\mathbf{x}_0,y)}\mathbb{E}_{q(\mathbf{x}_i|\mathbf{x}_{i+1},\mathbf{x}_0,y)}[\log \frac{p_\theta(\mathbf{x}_i|\mathbf{x}_{i+1},y)}{q(\mathbf{x}_i|\mathbf{x}_{i+1},\mathbf{x}_0,y)}] - D_{KL}(q(\mathbf{x}_T|\mathbf{x}_0,y)\|p_\theta(\mathbf{x}_T|y))\\
&= C - \sum_{i=1}^{T-1}\mathbb{E}_{q(\mathbf{x}_{i+1}|\mathbf{x}_0,y)}[D_{KL}(q(\mathbf{x}_i|\mathbf{x}_{i+1},\mathbf{x}_0,y)\|p_\theta(\mathbf{x}_i|\mathbf{x}_{i+1},y))]\\
&= -\mathbb{E}_{\boldsymbol{\epsilon},t}\left[w_t\|\boldsymbol{\epsilon}_\theta(\mathbf{x}_t,t,y)-\boldsymbol{\epsilon}\|_2^2\right] + C.
\end{aligned}
$$

We get the result of Eq. (5).

### A.4 CONNECTION BETWEEN ENERGY-BASED MODELS (EBMS)

The EBMs (LeCun et al., 2006) directly use neural networks to learn $p_\theta(\mathbf{x})$ and $p_\theta(\mathbf{x}|y)$.

$$
p_\theta(\mathbf{x}|y) = \frac{\exp(-E_\theta(\mathbf{x})_y)}{Z(\theta,y)},
$$

Where $E_\theta(\mathbf{x}) : R^D \rightarrow R^n$, and $Z(\theta,y) = \int \exp(-E_\theta(\mathbf{x})_y)d\mathbf{x}$ is the normalizing constant.

As described in Grathwohl et al. (2019), we can use EBMs to classify images by calculating the conditional probability:

$$
p_\theta(y|\mathbf{x}) = \frac{\exp(-E_\theta(\mathbf{x})_y)}{\sum_{\hat{y}} \exp(-E_\theta(\mathbf{x})_{\hat{y}})}. \tag{11}
$$

Compare Eq. (11) and Eq. (6), we can also set the energy function as:

$$
E_\theta(\mathbf{x})_y \approx \mathbb{E}_{t,\boldsymbol{\epsilon}}\left[w_t\|\boldsymbol{\epsilon}_\theta(\mathbf{x}_t,t,y)-\boldsymbol{\epsilon}\|_2^2\right]. \tag{12}
$$

Therefore, our diffusion classifier could be viewed as an EBM, and the energy function is the conditional diffusion loss.

## A.5 COMPUTING GRADIENT WITHOUT COMPUTING UNET JACOBI

We propose another way to compute the gradient of Eq. (9) without backpropagating through the UNet. Note that we do not use this method in any of the experiments. We only derive this method and conduct some theoretical analysis.

**Lemma A.1.** *Assuming that* $\mathbf{z} \sim \mathcal{N}(\boldsymbol{\mu}, \boldsymbol{\Sigma})$, $f : R^{n_i} \to R^{n_o}$, $p : R^n \to R$. *We can get*

$$\nabla_{\boldsymbol{\mu}} \mathbb{E}_{\mathbf{z}}[f(\mathbf{z})] = \mathbb{E}_{\mathbf{z}}[\nabla_{\boldsymbol{\mu}} \log p(\mathbf{z}) f(\mathbf{z})^T]. \tag{13}$$

*Proof.* Inspired by Wierstra et al. (2014), we derive

$$
\begin{aligned}
\nabla_{\boldsymbol{\mu}} E[f(\mathbf{z})] &= \nabla_{\boldsymbol{\mu}} \int f(\mathbf{z}) p(\mathbf{z}|\boldsymbol{\mu}) d\mathbf{z} \\
&= \lim_{d\boldsymbol{\mu} \to \mathbf{0}} \frac{\int f(\mathbf{z}) p(\mathbf{z}|\boldsymbol{\mu} + d\boldsymbol{\mu}) d\mathbf{z} - \int f(\mathbf{z}) p(\mathbf{z}|\boldsymbol{\mu}) d\mathbf{z}}{d\boldsymbol{\mu}} \\
&= \int \nabla_{\boldsymbol{\mu}} p(\mathbf{z}|\boldsymbol{\mu}) f(\mathbf{z})^T d\mathbf{z} \\
&= \int p(\mathbf{z}) \nabla_{\boldsymbol{\mu}} \log p(\mathbf{z}|\boldsymbol{\mu}) f(\mathbf{z})^T d\mathbf{z} \\
&= \mathbb{E}_z[\nabla_{\boldsymbol{\mu}} \log p(\mathbf{z}|\boldsymbol{\mu}) f(\mathbf{z})^T].
\end{aligned}
$$

$\square$

According to Lemma A.1, we can derive the gradient of Eq. (9) as

$$
\begin{aligned}
&\frac{d}{d\mathbf{x}} \mathbb{E}_{\boldsymbol{\epsilon}}[\|\boldsymbol{\epsilon}_\theta(\mathbf{x}_t, t) - \boldsymbol{\epsilon}\|_2^2] \\
=&\frac{d}{d\mathbf{x}} \mathbb{E}_{\mathbf{x}_t}[\|\boldsymbol{\epsilon}_\theta(\mathbf{x}_t, t) - \frac{\mathbf{x}_t - \sqrt{\alpha_t}\mathbf{x}}{\sigma_t}\|_2^2] \\
=&\frac{d}{d\mathbf{x}} \mathbb{E}_{\mathbf{x}_t}[g(\mathbf{x}_t, \mathbf{x}, t)] \\
=&\frac{\partial}{\partial \mathbf{x}_t} \mathbb{E}_{\mathbf{x}_t}[g(\mathbf{x}_t, \mathbf{x}, t)] \frac{\partial \mathbf{x}_t}{\partial \mathbf{x}} + \frac{\partial}{\partial \mathbf{x}} \mathbb{E}_{\mathbf{x}_t}[g(\mathbf{x}_t, \mathbf{x}, t)] \\
=&\mathbb{E}_{\mathbf{x}_t}[\frac{\partial \log p(\mathbf{x}_t|\mathbf{x})}{\partial \mathbf{x}} g(\mathbf{x}_t, \mathbf{x}, t)] + \frac{\partial}{\partial \mathbf{x}} \mathbb{E}_{\mathbf{x}_t}[g(\mathbf{x}_t, \mathbf{x}, t)] \\
=&\mathbb{E}_{\mathbf{x}_t}[\frac{\partial \log p(\mathbf{x}_t|\mathbf{x})}{\partial \mathbf{x}} \|\boldsymbol{\epsilon}_\theta(\mathbf{x}_t, t) - \frac{\mathbf{x}_t - \sqrt{\alpha_t}\mathbf{x}}{\sigma_t}\|_2^2] + \mathbb{E}_{\mathbf{x}_t}[2(\boldsymbol{\epsilon}_\theta(\mathbf{x}_t, t) - \frac{\mathbf{x}_t - \sqrt{\alpha_t}\mathbf{x}}{\sigma_t}) \frac{\sqrt{\alpha_t}}{\sigma_t}] \\
=&\mathbb{E}_{\boldsymbol{\epsilon}}[\frac{\partial \log p(\mathbf{x}_t|\mathbf{x})}{\partial \mathbf{x}} \|\boldsymbol{\epsilon}_\theta(\mathbf{x}_t, t) - \boldsymbol{\epsilon}\|_2^2] + \mathbb{E}_{\boldsymbol{\epsilon}}[(\boldsymbol{\epsilon}_\theta(\mathbf{x}_t, t) - \boldsymbol{\epsilon}) \frac{2\sqrt{\alpha_t}}{\sigma_t}].
\end{aligned}
\tag{14}
$$

Similarly, we can get the gradient of conditional diffusion loss

$$
\begin{aligned}
&\frac{d}{d\mathbf{x}} \mathbb{E}_{\boldsymbol{\epsilon}}[\|\boldsymbol{\epsilon}_\theta(\mathbf{x}_t, t, y) - \boldsymbol{\epsilon}\|_2^2] \\
=&\mathbb{E}_{\mathbf{x}_t}[\frac{\partial \log p(\mathbf{x}_t|\mathbf{x})}{\partial \mathbf{x}} \|\boldsymbol{\epsilon}_\theta(\mathbf{x}_t, t, y) - \frac{\mathbf{x}_t - \sqrt{\alpha_t}\mathbf{x}}{\sigma_t}\|_2^2] + \mathbb{E}_{\mathbf{x}_t}[(\boldsymbol{\epsilon}_\theta(\mathbf{x}_t, t, y) - \frac{\mathbf{x}_t - \sqrt{\alpha_t}\mathbf{x}}{\sigma_t}) \frac{2\sqrt{\alpha_t}}{\sigma_t}] \\
=&\mathbb{E}_{\boldsymbol{\epsilon}}[\frac{\partial \log p(\mathbf{x}_t|\mathbf{x})}{\partial \mathbf{x}} \|\boldsymbol{\epsilon}_\theta(\mathbf{x}_t, t, y) - \boldsymbol{\epsilon}\|_2^2] + \mathbb{E}_{\boldsymbol{\epsilon}}[(\boldsymbol{\epsilon}_\theta(\mathbf{x}_t, t, y) - \boldsymbol{\epsilon}) \frac{2\sqrt{\alpha_t}}{\sigma_t}].
\end{aligned}
\tag{15}
$$

As shown, the gradient of Eq. (9) have two terms. The first term equals to the weighted sum of $\frac{\partial \log p(\mathbf{x}_t|\mathbf{x})}{\partial \mathbf{x}}$. In VE-SDE case, where $\mathbf{x}_t = \mathbf{x} + \sigma_t \boldsymbol{\epsilon}$, the negative gradient direction is aligned with $\mathbf{x} - \mathbf{x}_t$ (a vector starting from $\mathbf{x}_t$ and ending at $\mathbf{x}$). The second term is proportional to the gradient of Score Distillation Sampling (Poole et al., 2022; Wang et al., 2023b), which also point toward real data. Consequently, optimizing the diffusion loss will move $\mathbf{x}$ toward a region with higher log likelihood.

**Algorithm 2** Training of multi-head diffusion
___
**Require:** A pre-trained diffusion model $\epsilon_\theta$, dataset $\mathcal{D}$, a multi-head diffusion model $\epsilon_\phi$
1:  **repeat**
2:    $\mathbf{x}_0, y \sim D$;
3:    $t \sim \text{Uniform}(\{1, 2, ..., T\})$, $\epsilon \sim \mathcal{N}(\mathbf{0}, \mathbf{I})$;
4:    **for** $y = 0$ **to** $K - 1$ **do**
5:      Take gradient descent step on $\nabla_\phi \mathbb{E}_{\epsilon, t}[w_t \| \epsilon_\theta(\hat{\mathbf{x}}_t, t, y) - \epsilon_\phi(\hat{\mathbf{x}}_t, t, y) \|_2^2]$;
6:    **end for**
7:  **until** converged;
___

# B  MORE EXPERIMENTAL RESULTS

## B.1  TRAINING DETAILS

**Computational resources.** We conduct Direct Attack on $1\times$ A40 GPUs due to the large memory cost of computational graphs for second-order derivatives. We use $2\times$ 3090 GPUs for other experiments. We also analyze the time complexity and test the real-time cost on a single 3090 GPU, as demonstrated in Table 3. We are unable to assess the real-time cost of some methods due to difficulties in replicating them.

**Training details of multi-head diffusion.** To reduce the time complexity of the diffusion classifier from $O(K \times T)$ to $O(T)$, we propose to slightly modify the architecture of the UNet, enabling it to predict for all classes at once. Since our changes are limited to the UNet architecture, all theorems and analyses remain applicable in this context.

However, this architecture only achieves 60% accuracy on the CIFAR10 dataset, even with nearly the same number of parameters as the original UNet. We tried to solve this problem by using a larger CFG (*i.e.*, viewing extrapolated result $(1 + \text{cfg}) \cdot \epsilon_\theta(\mathbf{x}_t, t, y) - \text{cfg} \cdot \epsilon_\theta(\mathbf{x}_t, t)$ as the prediction of UNet), but it does not work.

We hypothesize that with the traditional conditional architecture, the UNet focuses on extracting features relevant to specific class labels, leading to a more accurate prediction of the conditional score. In contrast, multi-head diffusion must extract features suitable for predicting all classes, as different heads use the same features for their predictions. To test this hypothesis, we measure the cosine similarity between features of a given $\mathbf{x}_t$ with different embeddings $y$. We find that for the traditional diffusion architecture, these features differ from each other. However, for multi-head diffusion, the cosine similarity of these predictions exceeds 0.98, indicating that the predictions are almost identical due to the identical feature.

It's worth noting that this does not mean traditional diffusion models are superior to multi-head diffusion. Both architectures have nearly the same number of parameters, as we only modify the last convolution layer. Additionally, the training loss curve and validation loss curve for both are almost identical, indicating they fit the training distribution and generalize to the data distribution similarly. The FID values of these two models are 3.14 and 3.13, very close to each other. The decreased performance of multi-head diffusion in the diffusion classifier is likely because it isn't clear on which feature to extract first. The training dynamic lets multi-head diffusion extracts features suitable for all classes, leading to similar predictions for each class, similar diffusion loss, and thus lower classification performance.

To prevent predictions for all classes from being too similar, we first considered training the multi-head diffusion with negative examples. Initially, we attempted to train the multi-head diffusion using the cross-entropy loss. While this achieved a training accuracy of 91.79%, the test accuracy only reached 82.48%. Moreover, as training continued, overfitting to the training set became more pronounced. **Notably, this model had 0% robustness.** Fortunately, this experiment underscores the strength of our adaptive attacks in evaluating such randomized defenses, affirming that the robustness of the diffusion classifier is not merely due to its stochastic nature leading to an inadequate evaluation. A lingering concern is our lack of understanding as to why switching the training loss from diffusion loss to cross-entropy loss drastically diminishes the generalization ability and robustness.

Our hypothesis posits that, when trained with the diffusion loss, diffusion models are compelled to extract robust features because they are required to denoise the noisy images. However, when

Table 3: Clean accuracy (%) and robust accuracy (%) of different methods against unseen threats.

| Method | Architecture | NFEs | Real Time (s) | Clean Acc | Robust Acc $\ell_\infty$ norm | $\ell_2$ norm | Avg |
|---|---|---|---|---|---|---|---|
| AT-DDPM-$\ell_\infty$ | WRN70-16 | 1 | 0.01 | 88.87 | 63.28 | 64.65 | 63.97 |
| AT-DDPM-$\ell_2$ | WRN70-16 | 1 | 0.01 | 93.16 | 49.41 | 81.05 | 65.23 |
| AT-EDM-$\ell_\infty$ | WRN70-16 | 1 | 0.01 | 93.36 | 70.90 | 69.73 | 70.32 |
| AT-EDM-$\ell_2$ | WRN70-16 | 1 | 0.01 | 95.90 | 53.32 | 84.77 | 69.05 |
| PAT-self | AlexNet | 1 | 0.01 | 75.59 | 47.07 | 64.06 | 55.57 |
| DiffPure ($t^* = 0.125$) | UNet | 126 | 0.72 | 87.50 | 53.12 | 75.59 | 64.36 |
| DiffPure ($t^* = 0.1$) | UNet | 101 | 0.60 | 90.97 | 53.52 | 72.65 | 63.08 |
| SBGC | UNet | $30TK$ | 15.78 | 95.04 | 0.00 | 0.00 | 0.00 |
| HybViT | ViT | 1 | 0.01 | 95.90 | 0.00 | 0.00 | 0.00 |
| JEM | WRN28-10 | 1 | 0.01 | 92.90 | 8.20 | 26.37 | 17.29 |
| Pérez et al. (2021) | WRN70-16 | 9 | n/a | 89.48 | 72.66 | 71.09 | 71.87 |
| Schwinn et al. (2022) | WRN70-16 | $KN$ | n/a | 90.77 | 71.00 | 72.87 | 71.94 |
| Blau et al. (2023) | WRN70-16 | $KN$ | n/a | 88.18 | 72.02 | 75.90 | 73.96 |
| LM (ours) | WRN70-16 | $1 + NT$ | 2.50 | 95.04 | 2.34 | 12.5 | 7.42 |
| LM (ours) | WRN70-16 | $1 + N$ | 0.10 | 87.89 | 71.68 | 75.00 | **73.34** |
| DC (ours) | UNet | $TK$ | 9.76 | 93.55 | 35.94 | 76.95 | **55.45** |
| RDC (ours) | UNet | $NT + TK$ | 12.26 | 93.16 | 73.24 | 80.27 | **76.76** |
| RDC (ours) | UNet | $N + TK$ | 9.86 | 88.18 | **80.07** | **84.76** | **82.42** |
| RDC (ours) | UNet | $N + T$ | 1.43 | 89.85 | 75.67 | 82.03 | **78.85** |

trained using the cross-entropy loss, there isn't a necessity to denoise the noisy images, so the models might not extract robust features. As a result, they may lose their image generation and denoising capabilities, as well as their generalization ability and robustness. We evaluated the diffusion loss of the diffusion models trained by cross-entropy loss and found that their diffusion losses hovered around 10. Furthermore, the images they generated resembled noise, meaning that they lose their generation ability.

To address this issue, we need to strike a balance between the diffusion loss, which ensures the robustness of the diffusion models, and the negative example loss (e.g., cross-entropy loss, CW loss, DLR loss) to prevent their predictions for various classes from becoming too similar. This balancing act turns the training of multi-head diffusion into a largely hyper-parameter tuning endeavor. To circumvent such a complex training process, we suggest distilling the multi-head diffusion from a pretrained traditional diffusion model. As illustrated in Algorithm 2, the primary distinction between multi-head diffusion distillation and traditional diffusion model training is that the predictions for all classes provided by the multi-head diffusion model are simultaneously aligned with those of a pre-trained diffusion model.

Note that in Algorithm 2, the predictions for different classes are computed in parallel. This approach sidesteps the need for tedious hyper-parameter tuning. Nevertheless, there's still potential for refinement. In this algorithm, the input pair $(\mathbf{x}_t, t, y)$ is not sampled based on its probability $p(\mathbf{x}_t, t, y) = \int p(\mathbf{x}|y)p(t)p(\mathbf{x}_t|\mathbf{x})p(y)d\mathbf{x}$. This could be why the multi-head diffusion slightly underperforms compared to the traditional diffusion model. Addressing this issue might involve using importance sampling, a potential avenue for future research.

## B.2 MORE ANALYSIS AND DISCUSSION

**Gradient magnitude.** When attacking the diffusion classifiers, we need to take the derivative of the diffusion loss. This is exactly what people do when training diffusion models, thus the gradient vanishing may not occur. We also measure the average absolute value of gradient (i.e., $\frac{1}{D}\|g\|_1$). As shown in Table 4, the mag-

Table 4: Gradient magnitudes of different methods

| Method | $\frac{1}{D}\|g\|_1$ |
|---|---|
| Engstrom et al. (2019) | $7.7 \times 10^{-6}$ |
| Wong et al. (2020) | $1.1 \times 10^{-5}$ |
| Salman et al. (2020) | $6.6 \times 10^{-6}$ |
| Debenedetti et al. (2022) | $9.8 \times 10^{-6}$ |
| Ours | $8.2 \times 10^{-6}$ |

nitude of the gradient of our method is at the same scale as that of other adversarial training models, which validates that our method does not have gradient vanishing.

**Substituting likelihood maximization with DiffPure.** We further study the performance by substituting likelihood maximization with DiffPure. We use the same hyperparameters as in Nie et al. (2022) and follow the identical evaluation setup as described in Sec. 4.1. The robustness of each method under the $\ell_\infty$-norm threat model with $\epsilon_\infty = 8/255$ on the CIFAR-10 dataset is shown in Table 5. As shown, DC+DiffPure outperforms DiffPure significantly, highlighting the effectiveness of our diffusion classifier. Furthermore, RDC surpasses DC+DiffPure, indicating that likelihood maximization is more compatible with the diffusion classifier. Besides, Xiao et al. (2023) provide an interesting explanation of DiffPure. It has been demonstrated that DiffPure increases the likelihood of inputs with high probability, resulting in better robustness. By directly maximizing the likelihood of inputs, our likelihood maximization further enhances the potential for improved robustness.

**Attacking using the adaptive attack in Sabour et al. (2015).** Tramer et al. (2020) proposes to add an additional feature loss (Sabour et al., 2015) that minimizes the class score between the current image and a target image in another class. This create adversarial examples whose class scores match those of clean examples but belong to a different class, thereby generating in-manifold adversarial examples, avoiding to be detected by likelihood-based adversarial example detectors.

Table 5: The robustness of DiffPure, DiffPure+DC and RDC.

| Method | Robustness(%) |
|---|---|
| DiffPure | 53.52 |
| DiffPure+DC | 69.92 |
| RDC | 75.67 |

To evaluate the robustness of our method against these adaptive attacks, we integrate them with AutoAttack and test the robust accuracy under $\ell_\infty$ threat model with $\epsilon_\infty = 8/255$. Surprisingly, our method achieves 90.04% robustness against attack using feature loss, and 86.72% robustness against attack using feature loss combined with the cross entropy loss or DLR loss in AutoAttack. On one hand, our Lagrange attack in Sec. 4.4 directly maximizes the lower bound of likelihood, making it more effective than feature loss. On the other hand, our method does not incorporate adversarial example detectors, making it unnecessary to strictly align the logits of adversarial examples with those of clean images.

**Comparison with other dynamic defenses.** We also compare our methods with state-of-the-art dynamic defenses. As some of these methods have not yet been open-sourced, we reference the best results reported in their respective papers. We use $N$ to denote the optimization steps in their methods (*e.g.*, qualification steps in Schwinn et al. (2022), PGD steps in Blau et al. (2023)). As shown in Table 3, our methods are not only more efficient but also effective than these dynamic defenses. Specifically, the time complexities of these dynamic defenses are related to the number of classes $K$, which limits their applicability in large datasets. On the contrary, the time complexity of our RDC does not depend on $K$. Moreover, our RDC outperforms previous methods by +3.01% on $\ell_\infty$ robustness and +6.33% on $\ell_2$ robustness, demonstrating the strong efficacy and efficiency of our RDC.

**Comparison with other randomized defenses.** As shown in Table 6, our method outperforms previous state-of-the-art randomized defenses. This is because diffusion models are naturally robust to such Gaussian corruptions, and such high variance Gaussian corruptions are much more effective than Fu et al. (2021); Dong et al. (2022) to smooth the local extrema in loss landscape, preventing the existence of adversarial examples.

Table 6: Comparison with other randomized defenses.

| Method | Attacker | Robustness(%) |
|---|---|---|
| Fu et al. (2021) | PGD-100 | 66.28 |
| Dong et al. (2022) | PGD-20 | 60.69 |
| Hao et al. (2022) | n/a | 0 |
| RDC (Ours) | AutoAttack | 75.67 |

**Comparison between different likelihood maximizations.** We compare the LM $(1 + NT)$ with the improved version LM $(1 + N)$. Surprisingly, under the BPDA attack, LM $(1 + NT)$ achieves only 2.34% robustness. On the one hand, the likelihood maximization moves the inputs towards high log-likelihood region estimated by diffusion models, instead of traditional classifiers, thus it is more effective when combined with diffusion classifiers. On the other hand, although the diffusion losses of LM $(1 + NT)$ and LM $(1 + N)$ are same in expectation, the former induces less randomness, thus it is less effective to smooth the local extrema. LLet's delve into a special case with $N = 1$. In this case, the expectation of LM $(1 + NT)$ is $\mathbb{E}_\epsilon[f(\mathbf{x} + \nabla_\mathbf{x}\mathbb{E}_t[w_t\|\epsilon_\theta(\mathbf{x}_t, t, y) - \epsilon\|_2^2])]$, while the expectation of LM $(1 + N)$ is $\mathbb{E}_{\epsilon,t}[f(\mathbf{x} + \nabla_\mathbf{x}[w_t\|\epsilon_\theta(\mathbf{x}_t, t, y) - \epsilon\|_2^2])]$. The primary difference between these two is the placement of the expectation over $T$ for LM $(1 + N)$, which is outside the function $f$. This arrangement implies that the randomness associated with $t$ also aids in smoothing out local extrema,

Table 7: Clean accuracy (%) and robust accuracy (%) of different methods in Restricted ImageNet.

| Method | Clean Acc | Robust Acc |
|---|---|---|
| Engstrom et al. (2019) | 87.11 | 53.12 |
| Wong et al. (2020) | 83.98 | 46.88 |
| Salman et al. (2020) | 86.72 | 56.64 |
| Debenedetti et al. (2022) | 80.08 | 38.67 |
| DiffPure (Nie et al., 2022) | 81.25 | 29.30 |
| RDC (ours) | **87.50** | **58.40** |

leading to better smoothed landscape and higher robustness. It is essential to clarify that this is not a result of the stochasticity hindering the evaluation of their robustness. We have already accounted for their stochasticity by applying EOT 100 times, as illustrated in Fig. 2(a).

**Comparison between different RDCs.** As shown in Table 3, our vanilla RDC attains 73.24% $\ell_\infty$ robustness and 80.27% $\ell_2$ robustness, surpassing prior adversarial training and diffusion-based purification techniques. By substituting the LM with the enhanced likelihood maximization, we manage to further boost the robustness by 6.83% and 4.49% against the $\ell_\infty$ and $\ell_2$ threat models, respectively. When employing multi-head diffusion, the RDC's time complexity significantly diminishes, yet its robustness and accuracy remain intact. This underscores the remarkable efficacy and efficiency of our proposed RDC.

### B.3 EXPERIMENT ON RESTRICTED IMAGENET

**Datasets and training details.** We conduct additional experiments on Restricted ImageNet (Tsipras et al., 2019), since Karras et al. (2022) provides off-the-shelf conditional diffusion model for imagenet dataset. Restricted ImageNet is a subset of ImageNet with 9 super-classes. For robustness evaluation, we randomly select 256 images from Restricted ImageNet test set due to the high computational cost of the attack algorithms, following Nie et al. (2022).

**Hyperparameters and robustness evaluation.** We use the same hyper-parameters and robustness evaluation as in Sec. 4.1. Following Nie et al. (2022), we only evaluate $\ell_\infty$ robustness with $\epsilon_\infty = 4/255$ in this subsection.

**Compared methods.** We compared our method with four state-of-the-art adversarial training models (Engstrom et al., 2019; Wong et al., 2020; Salman et al., 2020; Debenedetti et al., 2022) and DiffPure (Nie et al., 2022). For discriminative classifiers such as adversarially trained models, DiffPure, and LM, we compute the logit for each super-class by averaging the logits of its associated classes. For our RDC, we select the logit of the first class within the super-class to stand for the whole super-class.

**Results.** As shown in Table 7, our RDC outperforms previous methods by +1.75%, even though RDC only uses the logit of the first class of each super class for classification. This demonstrates that our method is effective on other datasets as well.

### B.4 EXPERIMENT ON CIFAR-100

We also test the robustness of different method against $\ell_\infty$ threat model with $\epsilon_\infty = 8/255$, following the same experimental settings as CIFAR-10. Due to the time limit, we only random sample 128 images. The results are shown in Table 8.

We find that RDC still achieves superior result compared with the state-of-the-art adversarially trained models and DiffPure. More surprisingly, we discover that DiffPure does not work well on CIFAR-100. We guess this is because CIFAR-100 has more fine-grained classes, and thus a small amount of noise will make the image lose its semantic information of a specific class. Hence, DiffPure is not suitable for datasets with more fine-grained classes but small resolution. This experiment indicate that our methods could be easily scaled to fine-grained datasets.

Table 8: Clean Accuracy (%) and robust accuracy (%) on CIFAR-100.

| Method | Clean Acc | Robust Acc |
|---|---|---|
| WRN40-2 | 78.13 | 0.00 |
| Rebuffi et al. (2021) | 63.56 | 34.64 |
| Wang et al. (2023a) | 75.22 | 42.67 |
| DiffPure | 39.06 | 7.81 |
| DC | 79.69 | 39.06 |
| RDC | **80.47** | **53.12** |

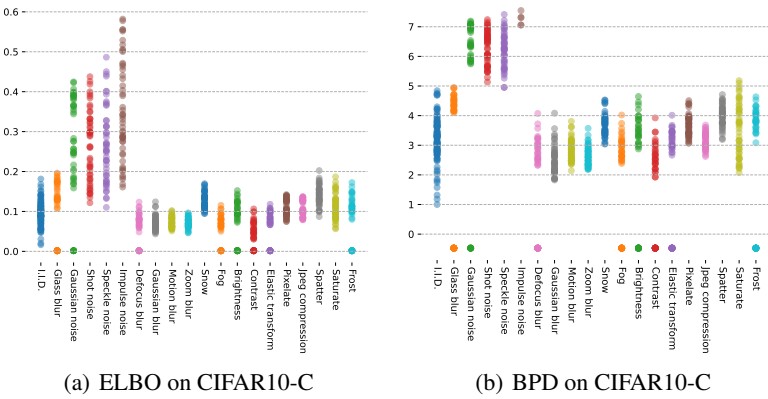

(a) ELBO on CIFAR10-C        (b) BPD on CIFAR10-C

Figure 3: The prediction of ELBO and BPD on CIFAR-10 test set and CIFAR-10-C.

### B.5 DISCUSSIONS.

**O.O.D. Detection.** We test both the unconditional ELBO and the likelihood (expressed in Bits Per Dim (BPD) as mentioned in Papamakarios et al. (2017)) on the CIFAR-10 test set and CIFAR10-C. As demonstrated in Fig. 3, while both methods could distinguish in-distribution data from certain types of corruptions such as Gaussian blur and Gaussian noise, they struggled to differentiate in-distribution data from corruptions like fog and frost.

**Generation of multi-head diffusion.** Since our multi-head diffusion is initialized from an unconditional EDM and distilled by a conditional EDM, it achieves a generative ability comparable to EDM. The images generated by our multi-head diffusion are shown in Fig. 4.

## C LIMITATIONS

Despite the great improvement, our methods could still be further improved. Currently, our methods requires $N + T$ NFEs for a single images, and applying more efficient diffusion generative models (Song et al., 2023; Shao et al., 2023; Liu et al., 2023) may further reduce $T$. Additionally, while we directly adopt off-the-shelf diffusion models from Karras et al. (2022), designing diffusion models specifically for classification may further improve performance. We hope our work serves as an encouraging step toward designing robust classifiers using generative models.

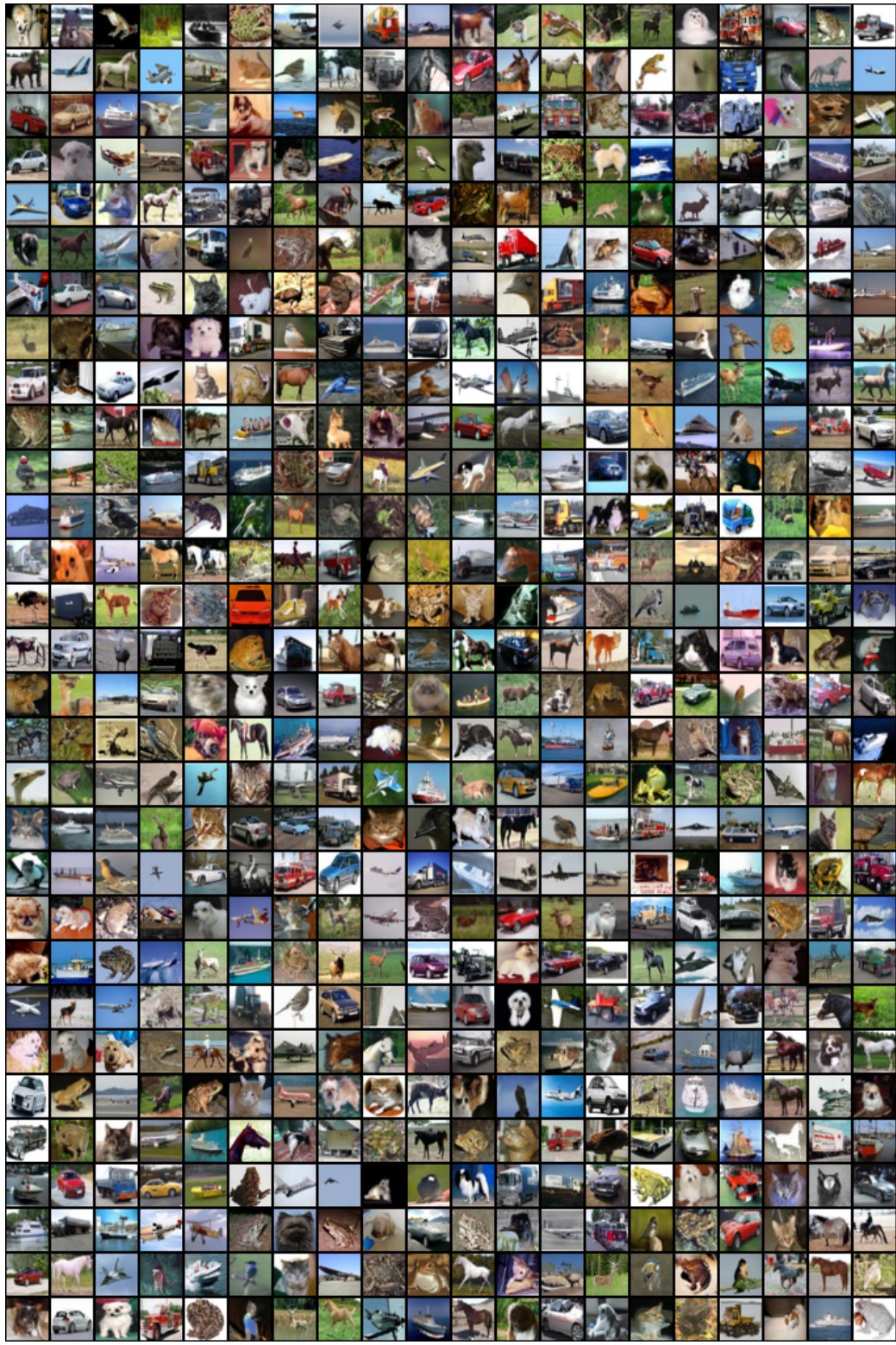

Figure 4: The images generated by multi-head diffusion.

