# OpenReview forum: "Robust Classification via a Single Diffusion Model"
_ICLR.cc/2024/Conference — Submitted to ICLR 2024_

### Official Review · Reviewer_noA3 · 2023-10-30

**Soundness:** 4 excellent
**Presentation:** 4 excellent
**Contribution:** 3 good
**Rating:** 8
**Confidence:** 3

**Summary:**

This work proposes a diffusion-based classifier that surpasses the SOTA level of robust accuracy against lp-bound adversarial attacks without relying on adversarial training. The method is advantageous to previous methods because it does not require inference for every class in the dataset by leveraging a "multi-head diffusion" block. Further, to improve the density estimation of real-world models the authors propose a "Likelihood Maximization" technique.

**Strengths:**

1. Clear and quality writing.
2. The method is carefully designed with theoretical justifications. In particular, the authors go to great lengths to address gradient obfuscation.
3. The method is benchmarked against modern adversarial attacks (AA, StAdv, BPDA) and beats SOTA by a large margin while showing generalization to more threats than previous methods.
4. Thorough review of related work and comparison to previous methods.

**Weaknesses:**

1. The manuscript mentions multiple times the expensive inference in both time and memory - however, a quantitative analysis is missing. I would like to see a table with inference time and memory for this method in comparison to other generative classifiers and regular ones.
2. The method appears to work well on "unseen" threats. However, all threats are limited to adversarial attacks. Is there any evidence for an increased robustness to other robustness aspects such as common corruptions (e.g., CIFAR10-C)?
3. The experimental evaluation is performed on a clearly separated and limited number of classes. Are there any results or theoretical insights on how this method would scale to more and potentially more fine-grained classes?

**Questions:**

1. In Appendix B the authors state "However, this architecture only achieves 60% accuracy on the CIFAR10 dataset". How does that relate to the 90-ish % in Tab. 1? I.e. what is different in this section?
2. “... [the mthod] ... leverages an off-the-shelf diffusion model” - how is this possible when the last layer is modified? Which diffusion model is this exactly? How does it compare to the diffusion models previous works have used?

---

> ### Author Response · Authors · 2023-11-17
> **Thank you for the valuable review (1/2)**
>
> Thank you for the supportive review. We are encouraged by the appreciation of the clarity, soundness, and effectiveness of this work. We have uploaded a revision of our paper. Below we address the detailed comments, and hope that you may find our response satisfactory.
>
>
>
> **Question 1: Qualitative analysis of time complexity in terms of real-time cost.**
>
> Thank you for your advice. We test the memory and real-time cost on a single 3090 GPU. The results are shown below:
>
>
> | Method     | NFEs | Real Time | Memory Cost (MB) |
> |:-----------|:----:|:---------:|:----------------:|
> | WRN-70-2   |  1   |   0.01    |       973        |
> | AT-DDPM    |  1   |   0.01    |       973        |
> | AT-EDM     |  1   |   0.01    |       973        |
> | JEM        |  1   |   0.01    |       1655       |
> | HybViT     |  1   |   0.01    |       2502       |
> | SBGC       | 30TK |   15.78   |       3557       |
> | DiffPure   | 101  |   0.60    |       3089       |
> | LM (ours)  |  T   |   0.10    |       3401       |
> | DC (ours)  |  TK  |   9.76    |       2664       |
> | RDC (ours) | N+TK |   9.86    |       3661       |
> | RDC (ours) | N+T  |   1.43    |       3661       |
>
>
> We have also supplemented this analysis in the Table 3 in Appendix. As shown, our multi-head diffusion greatly improves the efficiency while maintains the accuracy and robustness.
>
>
>
> **Question 2: Experiments on CIFAR10-C.**
>
> Thank you for your advice. The results on CIFAR10-C are shown below.
>
>
> |                   | WRN70-2  | AT-DDPM | AT-EDM | DiffPure| DC   |
> | :---------------- | :-------: | :-------------------: | :----------------: | :------: | :-------: |
> | Glass blur        | 41\.2     | 52\.2                 | **55\.4**          | 54\.0    | 36\.4     |
> | Gaussian noise    | 45\.8     | 69\.6                 | **73\.0**          | 72\.6    | 54\.0     |
> | Shot noise        | 65\.4     | 87\.0                 | **90\.0**          | 89\.0    | 69\.0     |
> | Speckle noise     | 69\.0     | 87\.2                 | **90\.8**          | 87\.8    | 67\.0     |
> | Impulse noise     | 50\.4     | 75\.0                 | **77\.4**          | 81\.4    | 33\.4     |
> | Defocus blur      | 59\.0     | 55\.6                 | **59\.6**          | 56\.0    | 58\.8     |
> | Gaussian blur     | 71\.6     | 83\.4                 | **89\.0**          | 84\.4    | 88\.2     |
> | Motion blur       | 84\.8     | 83\.2                 | **88\.0**          | 83\.4    | 85\.8     |
> | Zoom blur         | 81\.8     | 81\.8                 | **90\.2**          | 84\.2    | 88\.6     |
> | Snow              | 91\.0     | 85\.4                 | **91\.2**          | 87\.8    | 86\.2     |
> | Fog               | **61\.6**     | 47\.8                 | 51\.6              | 47\.0    | 58\.4 |
> | Brightness        | **62\.6**     | 57\.2                 | 60\.2              | 58\.4    | 60\.6 |
> | Contrast          | 57\.9     | 37\.2                 | 42\.0              | 42\.8    | **58\.0** |
> | Elastic transform | 58\.0     | 54\.4                 | **59\.2**          | 54\.6    | 56\.2     |
> | Pixelate          | 82\.2     | 89\.4                 | **92\.2**          | 87\.4    | 83\.0     |
> | JPEG compression  | 83\.8     | 88\.8                 | **92\.6**          | 88\.0    | 87\.2     |
> | Spatter           | 90\.8     | 86\.8                 | **91\.6**          | 86\.6    | 86\.8     |
> | Saturate          | **95\.0** | 86\.8                 | 91\.0              | 88\.2    | 91\.4     |
> | Frost             | 57\.8     | 52\.8                 | 57\.8              | 56\.6    | **58\.2** |
> | Average           | 69\.0     | 71\.2                 | 75\.9              | 73\.1    | 69\.4     |
>
>
>
> The performance of our method does not exceed the baselines. We observe that our diffusion classifier assigns extremely low likelihood to out-of-distribution (OOD) samples. This may be because our diffusion classifier accurately captures the distribution of the training set but does not generalize well to data that is outside of this distribution. We intend to investigate this issue further and aim to enhance the OOD accuracy in future work.

---

> > ### Comment · Reviewer_noA3 · 2023-11-18
> > **Thank you for your rebuttal**
> >
> > Thank you for your rebuttal.
> >
> > 1. Could you clarify why there are 2 AT-DDPM columns? Is it $\ell_\infty/_2$?
> > 2. Could you clarify what you mean by [low-likelihood on] "OOD" samples? Does that include adversarial samples or only CC samples?
> > 3. If your method assigns a low likelihood for OOD, it could be used as a detector. On ImageNet this could be tested on ImageNet-O [1]. This is out-of-scope for this paper and just a suggestion for future work. However, having a plot of the likelihood distribution for ID and OOD might enhance.
> >
> > [1] Hendrycks et al., "Natural Adversarial Examples", CVPR 2021.

---

> > > ### Author Response · Authors · 2023-11-22
> > > **Thank you for your follow-up comments.**
> > >
> > > Sorry for the late response. In the past few days, we have diligently worked on exploring out-of-distribution (OOD) detection and classification, following your advice. Below, we address the specific comments. We have also revised our submission considering your suggestions.
> > >
> > > ***Question 1: Incorrect table header.***
> > >
> > > We apologize for the error. The first column should be WRN70-2 (standard trained). This has been corrected in the revised official comment. The AT-DDPM model is trained with $\ell_\infty$ norm.
> > >
> > >
> > > ***Question 2: The definition of OOD in this context.***
> > >
> > > Precisely defining out-of-distribution (OOD) samples in this context is a complex issue that necessitates further investigation. In our view, each model estimates its unique distribution. A diffusion model constructs an estimated distribution from the training set, just as CNNs do. However, it appears that the distribution estimated by CNNs differs from that of diffusion models. For instance, most adversarial examples are considered OOD for CNNs, as they typically produce a large logit sum (Grosse et al., 2017), but this may not be the case for diffusion models. The latter may not regard these examples as OOD due to their limited impact on the diffusion classifier, which has an architecture markedly different from CNNs and thus poor transferability.
> > >
> > > Our observations indicate that diffusion models, when employing likelihood, can detect certain types of OOD examples, including categories like Gaussian blur and Gaussian noise, but not others such as fog, frost, and adversarial examples targeting CNNs, as shown in Appendix B.5 of our revision. This suggests that leveraging diffusion models for outlier detection merits additional exploration and refinement.
> > >
> > >
> > >
> > > ***Question 3: Inclusion of likelihood plots for in-distribution and OOD examples.***
> > >
> > > Thank you for your suggestion. We have included two figures and provided additional analysis in Appendix B.5.
> > >
> > >
> > >
> > > ***Question 4: OOD detection on ImageNet-O.***
> > >
> > >
> > > We conduct an experiment that use ELBO as confidence for OOD detection on ImageNet-O. We compare our algorithm with three previous methods mentioned in Hendrycks et al. (2021). The results are shown below:
> > >
> > > | Method       | AUPR ($\uparrow$) |
> > > |:-------------|:----:|
> > > | ResNet       | 14.3 |
> > > | DenseNet-121 | 15.8 |
> > > | ResNeXt-101  | 20.6 |
> > > | Hendrycks et al. (2019) | 16.2 |
> > > | ELBO (ours)  | 50.3 |
> > >
> > > We argue that using the likelihood for OOD example detection may warrant further investigation. Chen et al. (2023) demonstrate that  the probability mass of an isotropic Gaussian distribution is concentrated on the sphere, not at its center, yet the center exhibits high likelihood. In a similar vein, the image with the highest likelihood in diffusion models is one where all pixel values are zero. Such images, while having high likelihood, are unlikely to be generated by diffusion models due to their low probability. Therefore, we suggest that likelihood may not be the most direct metric for outlier detection. Instead, what might be more effective is calculating $p(o|\textbf{x})$, i.e., the probability of an image $\textbf{x}$ be outliers. A possible approach could involve using Bayes' theorem to establish a relationship between $p(o|\textbf{x})$ and likelihood, such as $p(o|\textbf{x})=1-\frac{p(\textbf{x}|\bar{o})p(\bar{o})}{p(\textbf{x})}$.
> > >
> > >
> > > Unfortunately, we will not be able to test this method before the end of the discussion phase. We apologize for the late response and our incomplete exploration of this issue. We plan to continue studying this problem in future work.
> > >
> > >
> > >
> > >
> > >
> > >
> > >
> > >
> > >
> > > ***
> > >
> > > **Reference:**
> > >
> > > Grosse, Kathrin, et al. "On the (statistical) detection of adversarial examples." arXiv preprint arXiv:1702.06280 (2017).
> > >
> > >
> > >
> > > Hendrycks, Dan, et al. "Natural adversarial examples." Proceedings of the IEEE/CVF Conference on Computer Vision and Pattern Recognition. 2021.
> > >
> > >
> > > Hendrycks, Dan, et al. "Using self-supervised learning can improve model robustness and uncertainty." Advances in neural information processing systems, 2019.
> > >
> > >
> > >
> > > Chen, Defang, et al. "A Geometric Perspective on Diffusion Models." arXiv preprint arXiv:2305.19947 (2023).

---

> ### Author Response · Authors · 2023-11-17
> **Thank you for the valuable review (2/2)**
>
> **Question 3: Scalability to dataset with more fine-grained classes.**
>
>
> Thank you for the suggestion. To verify the scalability of our method to fine-grained dataset, We test the robustness of different methods against $\ell_\infty$-norm threat model with $\epsilon_\infty=8/255$ on the CIFAR-100 dataset, following the same experimental settings as CIFAR-10. Due to the time limit, we only random sample 128 images for evaluation. The results are shown below.
>
>
> | Method | Clean Acc | Robust Acc|
> |---------|:-------:|:--------:|
> | WRN-40-2 |   78.13     | 0.00 |
> | AT-DDPM |    63.56     |   34.64   |
> | AT-EDM|  75.22        |   42.67 |
> | DiffPure  |         39.06         |     7.81     |
> | DC  |    79.69    |  39.06|
> |RDC |      80.47       |  53.12|
>
>
>
>
> We find that RDC still achieves superior result compared with the state-of-the-art adversarially trained models and DiffPure.
> More surprisingly, we discover that DiffPure does not work well on CIFAR-100. We guess this is because CIFAR-100 has more fine-grained classes, and thus a small amount of noise will make the image lose its semantic information of a specific class. Hence, DiffPure is not suitable for datasets with more fine-grained classes and small resolution. This experiment indicates that our methods could be scaled to datasets with more fine-grained classes.
>
>
>
>
>
>
> **Question 4: Training of multi-head diffusion.**
>
> As detailed in Appendix B.2, training a multi-head diffusion model directly results in an accuracy of 60\%. To address this, we experimented with introducing negative examples and knowledge distillation. We discovered that initializing the model from an unconditional diffusion model and then refining it through distillation with a conditional diffusion model (as outlined in Algorithm 2) effectively resolves the problem.
>
>
>
> **Question 5: Clarification of usage of the off-the-shelf diffusion model.**
>
>
> We apologize for the ambiguity. In this paper, we initially explore the robustness of a generative classifier using a single off-the-shelf model. Subsequently, we propose the multi-head diffusion, which requires training and is not an off-the-shelf model. We have revised this in the revision.

---

### Official Review · Reviewer_ScPt · 2023-10-30

**Soundness:** 4 excellent
**Presentation:** 3 good
**Contribution:** 4 excellent
**Rating:** 8
**Confidence:** 4

**Summary:**

This paper presents the Robust Diffusion Classifier (RDC), a generative classifier designed for robust classification tasks. RDC operates by first optimizing the data likelihood of an input and then estimating class probabilities for this optimized input. This is achieved using the diffusion model with transforms using the Bayes' rule. Recognizing the need for classification efficiency, the authors introduce a novel diffusion backbone termed "multi-head diffusion" and for a sampling strategy with fewer NFE. Notably, the method requires no specific training against particular adversarial attacks, showcasing its adaptability in defending against a spectrum of previously unseen threats.

**Strengths:**

- The proposed methodology stands out as both technically sound and effective. Its capability to achieve robust classification without specific knowledge of adversarial attacks is admirable. The paper has both theoretical and empirical contributions, which are beneficial to both the theoretical-favored researchers and practitioners.

- The paper is overall well-written and provides a smooth reading experience. The incorporation of model overviews and illustrative diagrams further clarifies the proposed methodology, enhancing comprehension.

- The experimental results well validate the approach. Notably, the paper also provides an ablation study to interpret the effects of the hyperparameters.

 - The authors have provided a thorough study that motivates the design of the multi-head classifier in the appendix. This study provides additional insight for researchers to figure out problems in related domains.

- I appreciate the authors' transparency in addressing potential limitations. The practical side of the research is solid. The authors have been very detailed in their implementation and provided their experiment code, which ensures reproducibility.

**Weaknesses:**

Although the content of the current version is satisfactory, some points listed below can further enhance the depth and completeness of this work:

- Some details regarding the diffusion model need clarification. For example, the sampling strategy is not very clear to me. It seems the authors deploy a VP sampling with uniform sliding timesteps similar to the one used in Nichol and Dhariwal, (2021). In the appendix and the code the authors also seem to leverage some implementation from Karras et al. (2022). As we note the sampler in Karras et al. (2022) involves additional correction steps, the NFE of the diffusion backbone would be more than T.

- The theoretical results are based on the hypothesis that the evidence lower bound is tight. It would be intriguing to explore the implications of the gap between the likelihood and this lower bound, especially when viewed through the lens of the Bayesian framework for uncertainty quantification. Exploring how this gap influences robust classification performance could enrich the paper's depth and utility.


- While the current experiments are limited to relatively small-scale datasets, there is inherent value in examining the method's scalability. It would be beneficial if the authors could present results or potential methodologies to apply their approach to larger datasets, drawing inspiration from other generative classifiers in the Bayesian paradigm, such as those highlighted by Heek and Nal (2019) and Han et al. (2022).


- This paper also has close relation to other generative classifiers not specified for robust classification. Although the authors have discussed some concurrent works, some prior works may need to include and discuss potential connections.

- The current form of the paper draws parallels to several generative classifiers, though not specifically designed for robust classification. While some works have been discussed as concurrent works, it might be helpful in enhancing the completeness to integrate and discuss other prior works, emphasizing their relevance and potential connections to the proposed methodology.

----------
Reference:

Heek, Jonathan, and Nal Kalchbrenner. "Bayesian inference for large scale image classification." arXiv preprint arXiv:1908.03491 (2019).

Mackowiak, R., Ardizzone, L., Kothe, U., & Rother, C. (2021). Generative classifiers as a basis for trustworthy image classification. In Proceedings of the IEEE/CVF Conference on Computer Vision and Pattern Recognition (pp. 2971-2981).

Hoogeboom, E., Nielsen, D., Jaini, P., Forré, P., & Welling, M. (2021). Argmax flows and multinomial diffusion: Learning categorical distributions. Advances in Neural Information Processing Systems, 34, 12454-12465.

Han, X., Zheng, H., & Zhou, M. (2022). Card: Classification and regression diffusion models. Advances in Neural Information Processing Systems, 35, 18100-18115.

**Questions:**

Please see the first point of the weakness regarding the details in diffusion settings.

---

> ### Author Response · Authors · 2023-11-17
> **Thank you for the valuable review**
>
> Thank you for the supportive review. We are encouraged by the appreciation of the soundness, clarity, and thoroughness of this work.  We have uploaded a revision of our paper. Below we address the detailed comments, and hope that you may find our response satisfactory.
>
> **Question 1: Clarification of the diffusion classifier.**
>
> We would like to clarify that the time complexity of our diffusion classifier is not related to the sampling complexity. Our diffusion classifier initially calculates the Evidence Lower Bound (ELBO) to approximate the log likelihood, and then computes the classification probability using $p(y|\textbf{x}) \propto p(\textbf{x}|y)$. Therefore, the time complexity of our method is determined by the time required to calculate the ELBO. For each class, we need $T$ NFEs to calculate its ELBO, hence we need $K\times T$ NFEs to calculate $p(y|\textbf{x})$.
>
> **Question 2: The gap between the ELBO and log likelihood.**
>
> We agree that exploring the gap between ELBO and log likelihood on the robustness of diffusion classifier is quite interesting. However, calculating the exact likelihood of  the diffusion model is extremely hard. Song et al. (2021) suggest using the instantaneous change of variable theorem (Chen et al., 2018) to compute the likelihood that a diffusion ODE generates certain images. However, this likelihood is not equivalent to the one in the evidence lower bound (ELBO), which actually represents the likelihood of the diffusion SDE generating such an image. These two likelihoods are only equal when the score function is optimal. In practice, we train a UNet to approximate the score of the real data distribution, which is certainly not optimal. Consequently, these two scores are not identical, and we have not found any method to calculate the exact likelihood of the diffusion SDE. We will strive to explore new ways to calculate the exact likelihood and the gap between ELBO and log likelihood in  future work.
>
> **Question 3: Potential solution to scale to large datasets.**
>
> Thank you for the suggestion. In Appendix B.3, we have provided an experiment on Restricted ImageNet, which shows the effectiveness of our approach to larger datasets. During the rebuttal phase, as suggested by Reviewer Xkzq, we supplement an experiment on CIFAR-100. The results of different methods are shown below.
>
> | Method | Clean Acc | Robust Acc|
> |---------|:-------:|:--------:|
> | WRN-40-2 |78.13| 0.00 |
> | AT-DDPM|63.56| 34.64|
> | AT-EDM  | 75.22 |42.67|
> | DiffPure | 39.06|7.81|
> | DC  |79.69|39.06|
> |RDC |80.47|53.12|
>
> Our proposed RDC achieves superior result compared with state-of-the-art adversarially trained models and DiffPure.
> These two experiments imply the scalability of our method to larger datasets with more fine-grained classes.
>
>
> We apologize for the confusion regarding the application of the methods proposed by Heek and Nal (2019) and Han et al. (2022) to enhance the scalability of our approach. Our method, being a generative classifier, utilizes the ELBO to approximate the log likelihood. We argue that the primary challenge in adapting our method to larger datasets is finding a more efficient way to compute the ELBO. The ATMC sampler proposed by Heek and Nal (2019) is designed to accelerate the training of Bayesian networks, while Han et al. (2022) focus on constructing a diffusion bridge for converting logits from discriminative classifiers into more accurate and calibrated probabilities. We reckon that these methods are quite different from our paradigm since they are discriminative approaches and hence their method could not be directly used to improve the scalability of our diffusion classifier.
>
> **Question 4,5: Lack of discussion of some popular generative classifiers.**
>
> Thank you for your suggestion. We have incorporated a discussion of more popular generative classifiers (Mackowiak et al., 2021; Hoogeboom et al., 2021; Han et al., 2022) into Section 2 of the revision.
>
> ***
>
> **References:**
>
> Song, Yang, et al. "Score-based generative modeling through stochastic differential equations." International Conference on Learning Representation, 2021.
>
> Chen, Ricky TQ, et al. "Neural ordinary differential equations." Advances in Neural Information Processing Systems, 2018.
>
> Heek, Jonathan, and Nal Kalchbrenner. "Bayesian inference for large scale image classification." arXiv preprint arXiv:1908.03491 (2019).
>
> Mackowiak, R., et al. "Generative classifiers as a basis for trustworthy image classification." IEEE/CVF Conference on Computer Vision and Pattern Recognition, 2021.
>
> Hoogeboom, E., et al. "Argmax flows and multinomial diffusion: Learning categorical distributions." Advances in Neural Information Processing Systems, 2021.
>
> Han, X., et al. "Card: Classification and regression diffusion models." Advances in Neural Information Processing Systems, 2022.

---

> > ### Comment · Reviewer_ScPt · 2023-11-22
> > **Response to the authors**
> >
> > I appreciate the author's response in addressing my questions and concerns. After reading the revision and all reviews, my concerns are addressed.
> >
> > I am also interested in the observation that the robust accuracy does not saturate at $T=2000$, as mentioned by my co-reviewer and the authors. It would be good if the authors could add the results in the final manuscript and involve relative analysis in exploring the limit of robust accuracy in terms of T (if possible).
> >
> > A minor point revision:
> >
> > The real-time column in Table 3 may need a unit (s, min, or h).
> >
> > I believe the paper can be valuable to the community and thus increase my score.

---

> > > ### Author Response · Authors · 2023-11-22
> > > **Thank you for your follow-up comments.**
> > >
> > > Thank you for your follow-up comments and for increasing the score. Below, we address your detailed comments. We have also revised our submission according to your suggestions.
> > >
> > > ***Question 1: Exploring robustness with higher $T$ and the limit of robustness.***
> > >
> > > We appreciate your advice. In response, we will investigate the limit of robustness at higher values of $T$ and include these results in the final manuscript.
> > >
> > > ***Question 2: Lack of measuring unit in Table 3.***
> > >
> > > Thank you for highlighting this oversight. The unit in Table 3 is seconds. We have added this information in the revised version of the manuscript.

---

### Official Review · Reviewer_Xkzq · 2023-10-31

**Soundness:** 3 good
**Presentation:** 2 fair
**Contribution:** 4 excellent
**Rating:** 8
**Confidence:** 4

**Summary:**

This paper proposes to build a robust classifier using a single diffusion model by calculating $p (y|x)$ via $p (x|y)$. The authors identify likelihood maximization as a key ingredient for ensuring the adversarial robustness of such models. To address the high computational complexity associated with this type of classifiers, the authors further introduce a multi-head U-Net and ablate on efficient sampling methods. Experiment results on a subset of CIFAR-10 using BPDA-AutoAttack show that the proposed method achieves SOTA clean and adversarial accuracy.

**Strengths:**

This paper proposes an interesting and relevant framework for robust classification. Given how fast diffusion models are improving in comparison to traditional discriminative robust classifiers, this work opens a new method of building robust models. It thus has a lot of potential for inspiring future research that builds even better robust models. Furthermore, the authors are careful with evaluating the proposed method with strong adaptive attacks, providing justifications for the proposed robustness estimation methods.

**Weaknesses:**

The main weaknesses are twofold: computational complexity and paper presentation.

### Computational Complexity

Even with the proposed multi-head U-Net and other complexity reduction measures, the computational complexity still seems to be high. While the likelihood maximization step only requires $N=5$ forward and backward passes, approximating $p(x|y)$ requires $T$ U-Net queries. That being said, I agree that this drawback can be left for future work.

### Paper Presentation

Many important details are missing from the discussion. Some of them can be found in the appendix, but they really should be in the main text. This is especially the case for Section 3.3.
- Theorem 3.2 discusses "optimal diffusion models". In what sense is the diffusion model optimal? The proof to this theorem clarifies that such a model minimizes the noise estimation error, but this should be in the main text.
- There is a softmax operation in Theorem 3.2, but the quantity on which it operates is a scalar (a norm square divided by some variance). What does softmax exactly mean here? Same for Corollary 3.3.
- "We find that the optimal diffusion classifier achieves 100% robust accuracy in both cases, validating our hypothesize that accurate density estimation of diffusion models facilitates robust classification." How was this found? Also, it should be "hypothesis", not "hypothesize". What is the main gap between an optimal diffusion classifier and an empirical diffusion classifier? Is it the limited amount of training data? Or is it how well the U-Net is optimized?
- Section 3.5 says, "instead of calculating the diffusion loss using all timesteps like Eq. (9), we only sample a single timestep". How is this time step sampled? Uniformly randomly?
- Section 3.5 also says, "(BPDA) approximates the gradient with an identity mapping". How exactly is the identity mapping applied? A pseudo-code or Python code explanation would be appreciated.
- It would be nice to have the experiment results from the CIFAR-100 dataset to have some diversity in the evaluation.

**Questions:**

- Since AutoAttack with BPDA is used for evaluation, is the Square Attack component of AutoAttack also included in the evaluation? Does Square find additional examples on top of the BPDA gradient-based attacks?
- In Figure 2a, why is the robust accuracy barely over 30%? What is the value of $T$ and $T'$ for the main results (Table 1)? Does the result get even better if $T'$ is larger than 1000? How large is $T$ during the training of the diffusion model? If we train a diffusion model with fewer time steps (i.e., discretize the trajectory into less than 1000 steps during training), should we expect the resulting classifier obtained via the proposed method to work well with a smaller $T'$?
- With multi-head diffusion, is it true that only the last layer receives the class condition signal? How does this affect the performance compared with injecting class conditioning into various locations in the U-Net? It would also be nice to see some generations from this diffusion model.

---

> ### Author Response · Authors · 2023-11-17
> **Thank you for the valuable review  (1/2)**
>
> Thank you for appreciating our new contributions as well as providing the valuable feedback. We have uploaded a revision of our paper. Below we address the detailed comments, and hope that you may find our response satisfactory.
>
>
>
> ***Question 1: Computational complexity.***
>
> We agree that the high computational complexity is a drawback of diffusion classifiers, and more generally generative classifiers. In our paper, we have made significant efforts to reduce the computational complexity by proposing multi-head UNet and various accelerated sampling strategies, as detailed in Section 3.5. Recently, we also made some progress for further reducing the time complexity of diffusion classifier. We found that a significant drop in robust accuracy, primarily caused by reducing $T'$ (i.e., number of sampling timesteps), is due to the large variance in diffusion loss. Owing to this substantial variance, more instances of diffusion loss must be calculated to achieve stable predictions. However, if we manage to reduce the variance of the diffusion loss (for example, by using the same $\textbf{x}_t$ for different classes), it becomes feasible to decrease $T'$ and thus achieves faster inference speeds. By implementing this approach, **we successfully reduce the time complexity of the diffusion classifier by 200 times while maintaining the accuracy of clean data.** In the future, we will continue our work with the diffusion classifier, striving to further reduce its time complexity without compromising either robust or clean accuracy.
>
>
>
> ***Question 2: Unclear definition of optimal diffusion model.***
>
> Thank you for pointing this out. Yes, the optimal diffusion model  has minimal noise estimation error. We have revised our paper to make this clearer.
>
>
>
> ***Question 3: Unclear definition of softmax operator.***
>
> Sorry for the ambiguity. The softmax function is operated over all data samples. We have revised our paper to make this clearer.
>
>
>
>
>
> **Question 4: The gap between optimal diffusion classifier and empirical diffusion classifier.**
>
>
> The primary difference between an optimal diffusion classifier and an empirical diffusion classifier is that the empirical diffusion classifier does not generalize well to the test data. In essence, the diffusion model fails to provide accurate score estimations for the noised versions of the test data. Conversely, an optimal diffusion model can yield accurate score estimations for test data. We empirically test the optimal diffusion classifier against AutoAttack, which achieves 100\% robustness. This result indicates that if a diffusion model can generalize  well to the test set, it would attain 100\% adversarial robustness.
> The gap between the optimal diffusion model and the empirical one could be reduced by using more training data or designing advanced training techniques, which would further improve the performance of diffusion classifier.
> In this paper, we focus on an alternative strategy (i.e., likelihood maximization) that moves the input $\mathbf{x}$ to $\hat{\mathbf{x}}$ that diffusion model could give a more accurate score estimation.
>
>
>
> **Question 5: Lack of sampling details in likelihood maximization.**
>
> Thank you for pointing this out. As you mentioned, we uniformly sample a timestep and then calculate the diffusion loss at that specific timestep. This process is used to approximate the expected diffusion loss across all timesteps. We have made this clearer in the revision.
>
>
>
>
> **Question 6: Specific details of BPDA**
>
> Thank you for pointing this out. Here is the python code of BPDA:
>
> ```
> class LikelihoodMaximization(torch.autograd.Function):
>     @staticmethod
>     def forward(ctx, x: Tensor):
>         x = likelihood_maximization(x.detach().clone())
>         return x
>
>     @staticmethod
>     def backward(ctx, upstream_grad):
>         return upstream_grad
> ```
>
> In the code, we do the forward pass normally, but in the backward pass, we directly return the upstream gradient. In other words, we view the Jacobian of likelihood maximization as an identity matrix.

---

> ### Author Response · Authors · 2023-11-17
> **Thank you for the valuable review (2/2)**
>
> **Question 7: Result on CIFAR-100.**
>
> Thank you for the suggestion. We further conduct experiments on CIFAR-100 and test the robustness of different methods against $\ell_\infty$-norm threat model with $\epsilon_\infty=8/255$, following the same experimental settings as CIFAR-10. Due to the time limit, we only random sample 128 images for evaluation. The results are shown below.
>
>
> | Method | Clean Acc | Robust Acc|
> |---------|:-------:|:--------:|
> | WRN-40-2 |   78.13     | 0.00 |
> | AT-DDPM |    63.56     |   34.64   |
> | AT-EDM  |  75.22        |   42.67 |
> | DiffPure   |         39.06         |    7.81    |
> | DC  |    79.69    |  39.06|
> |RDC |      80.47       |  53.12|
>
>
>
>
>
> We find that RDC still achieves superior result compared with the state-of-the-art adversarially trained models and DiffPure.
> More surprisingly, we discover that DiffPure does not work well on CIFAR-100. We guess this is because CIFAR-100 has more fine-grained classes, and thus a small amount of noise will make the image lose its semantic information of a specific class. Hence, DiffPure is not suitable for datasets with more fine-grained classes and small resolution.
>
>
>
>
>
> **Question 8: Square attacks can not find more adversarial examples.**
>
> Yes, we used Square Attack in AutoAttack for robustness evaluation. We found that Square Attack can not find additional examples on top of the BPDA adaptive attack.
>
>
>
>
> **Question 9: Experiments about $T'$.**
>
> In Figure 2a, we show the robustness of Diffusion Classifier (DC), but not the Robust Diffusion Classifier (RDC). DC can only achieve 35.94\% robustness against the $\ell_\infty$ threat model with $\epsilon=8/255$, as shown in Table 1.
>
> During training, $T$ is set as 1000 in DDPM (Ho et al., 2020). EDM (Karras et al., 2022) extends the discrete diffusion model into continuous diffusion model, where $t$ represents the standard deviation of isotropic Gaussian noises added to images. Consequently, in the context of EDM, there is no equivalent concept of $T$. Instead, during training, a $t$ value is randomly sampled from the range $[\sigma_{min}, \sigma_{max}]$.
> $T'$ is set as 1000 in Table 1.
>
>
> As argued in Karras et al. (2022), the training and inference of diffusion model can be decoupled. During training, a continuous $t$ could provide more accurate score estimation for different $t$, or even play a role of regularization, making the diffusion model generalize better. Therefore, $T'$ during inference is not related with $T$ during training.
> Therefore, we assert that reducing $T$ during training does not contribute positively to the performance of the diffusion classifier. We believe that a significant drop in robust accuracy, primarily caused by reducing $T'$, is due to the large variance in diffusion loss (See Question 1 for a solution).
>
>
>
>
> We agree that adopting a large $T'$ may further improve the robustness of the diffusion classifier. We conduct another experiment where we set $T'=2000$. As shown below, the robustness is further improved.
>
>
> | $T'$ |   Robustness  |
> |---------|:-------:|
> |  1000   |    35.94    |
> | 2000    |    39.06       |
>
>
>
>
> **Question 10: Details about multi-head diffusion.**
>
> We did not add any class condition signal to the multi-head diffusion. The last layer of multi-head diffusion directly maps the feature into $3\times K \times H \times D$ dimension to output predictions for $K$ classes. If we inject the class condition signal into some intermediate locations in UNet, then we have to calculate $K$ times for the rest part of the UNet, because the activations of each class are not same after the injection.
>
>
> Thank you for your advice on including the generation of multi-head diffusion. We were surprised to find that our multi-head diffusion achieves a 1.96 FID on the CIFAR-10 dataset, demonstrating its generative ability, which is comparable to the state-of-the-art generative models.
>
>
>
> ****
>
>
>
>
>
> **References:**
>
> Ho, Jonathan, Ajay Jain, and Pieter Abbeel. "Denoising diffusion probabilistic models." Advances in Neural Information Processing Systems, 2020.
>
>
> Karras, Tero, et al. "Elucidating the design space of diffusion-based generative models." Advances in Neural Information Processing Systems 35 (2022): 26565-26577.
>
>
> Nie, Weili, et al. "Diffusion models for adversarial purification." International Conference on Machine Learning, 2022.

---

> > ### Comment · Reviewer_Xkzq · 2023-11-19
> > **Thanks for the response.**
> >
> > Thank you for the response. The response addressed most of my concerns, so I have increased the rating from 6 to 8.
> >
> > Just a couple of minor things:
> > - Is the softmax operator in Corollary 3.3 defined as the softmax across all classes?
> > - There are some occasions where the equations exceed the page margin. Please fix them.
> > - Please explain the four legend entries of Figure 2a in its caption, since I don't think they are used elsewhere. It should also be clarified in the caption that the figure was made on DC, not RDC.
> > - It would be nice to add $T' = 2000$ (and potentially even higher if computation allows) to Figure 2a to show whether the robust accuracy "saturates", which seems to be the case based on the number for $T' = 2000$.
> > - It's odd that Figure 2c is mentioned first in the text, then 2b, and finally 2a. It might make sense to change the order.
> > - It would be interesting to show some of the generated images from the diffusion models as figures in the paper.

---

> > > ### Author Response · Authors · 2023-11-20
> > > **Thank you for your follow-up comments.**
> > >
> > > Thank you for your follow-up comments and increasing the score. Below we address the detailed comments. We have further provided a revision considering your suggestions.
> > >
> > >
> > > ***Question 1: Softmax in Corollary 3.3.***
> > >
> > > Yes, the softmax operator in Corollary 3.3 is defined as the softmax across all classes. As demonstrated in Appendix A.1, $p(y|\textbf{x})$ is the $y$-th output of the softmax over the log likelihoods of different classes. Our diffusion classifier $f(\textbf{x})_y$ utilizes the evidence lower bound to approximate the log likelihood of class $y$, therefore, the softmax here refers to the softmax applied across all classes.
> > >
> > >
> > > ***Question 2: Equations exceed the page margin.***
> > >
> > > Thank you for pointing this out. We have fixed it in the revision.
> > >
> > >
> > >
> > > ***Question 3: Figure 2a's description.***
> > >
> > > Thank you for pointing this out. We have added descriptions of the legends and clarified that the experiment was performed on DC in the revision.
> > >
> > >
> > >
> > >
> > > ***Question 4: Experiments on higher $T$.***
> > >
> > > Thank you for your advice. We discover that the robust accuracy does not "saturates" at $T=2000$. We are conducting experiments with higher value of $T$, but it seems that we will not be able to finish the experiments before the end of the discussion phase. We will supplement the result in the final version.
> > >
> > >
> > >
> > >
> > > ***Question 5: The order of the figures.***
> > >
> > > Thank you for your advice. We have re-aranged the order of Figure 2a, 2b and 2c.
> > >
> > >
> > > ***Question 6: Adding images generated by multi-head diffusion.***
> > >
> > > Thank you for your suggestion. We have included several images in Appendix B.5.

---

### Meta-Review · Area_Chair_WX8w · 2023-12-08

**Metareview:**

The authors propose a diffusion model based classifier and evaluate its robustness on several benchmark datasets using standard adversarial attack algorithms. The reviewers raised some issues with the paper that were well addressed during the rebuttal discussion. In summary, the reviewers made the following positive points about the paper:
1. The paper leverages diffusion models to build robust classifiers, and given that diffusion models are constantly improving on large-scale pretraining datasets, the improvements are likely to translate to the framework built by the authors.
2. The authors demonstrate improvements relative to SOTA results on adversarial robustness for image classifications on several benchmark datasets, evaluating robustness based on BPDA-driven auto-attack.
3. The paper is in general well-written and the experiments are thorough.

However, as the AC, I find that there are still some serious unaddressed issues with the paper:
1. The theoretical statements do not seem linked to the robustness claims at all. The authors derive the optimal diffusion classifier, and simply state that "Empirically, we evaluate the robust accuracy of the optimal diffusion classifier under the ℓ∞ norm
with ϵ∞ = 8/255 and the ℓ2 norm with ϵ2 = 0.5", without mentioning how this evaluation was done and whether it is reasonable.
2. The complexity of the diffusion model makes it infeasible for the authors to run SOTA white-box gradient based attacks on their diffusion model. There have been several instances of papers claiming robustness results in the adversarial robustness literature that were later broken by stronger attacks, and while I appreciate the authors' efforts to design strong computationally efficient attacks, it is unsatisfactory that the authors were unable to run multi-step white-box gradient based attacks on their model (which have been shown to be reliably stronger), even for a smaller scale dataset. The only evidence presented by the authors in this regard is comparing the BPDA attack against a single step white-box gradient based attack, which is insufficient as prior work has demonstrated the need for multi-step attacks to compute the strongest adversarial examples (for example https://proceedings.mlr.press/v119/croce20b/croce20b.pdf uses 1000 steps to evaluate the effectiveness of the AutoPGD attack).
Since the authors propose a robustness mechanism for which strong white box attacks become computationally expensive, the onus remains on the authors to demonstrate that the claimed robustness improvements hold against the strongest white-box attacks.

The seriousness of these two issues makes it hard to determine whether the results in the paper are true improvements in adversarial robustness that will persist against the strongest white box attacks.

Hence, I recommend rejection for now despite the reviewer assessments, but encourage the authors to resubmit the work to a future venue after evaluating the robustness of their approach against strong white box attacks, at least on smaller datasets where this is computationally feasible.

**Justification For Why Not Higher Score:**

Theoretical claims do not justify the robustness results, and the empirical evaluations, while seemingly comprehensive, do not include evaluation against the strongest gradient based adversarial attacks. Given the evidence available, it is difficult to judge whether the authors have computed truly optimal attacks against this model, bringing the claims of adversarial robustness into question.

**Justification For Why Not Lower Score:**

N/A

---

> ### Public Comment · ~Ziruo_Wang1 · 2024-04-02
> **Factural error in meta review**
>
> Dear AC,
>
> You completely misunderstood the evaluation of this paper.
>
> The AC stated that “the authors were unable to run multi-step white-box gradient-based attacks”. However, this is a factual error. As stated in Section 4.1, the authors adopted AutoAttack for robustness evaluation, which is exactly the strongest method mentioned by the AC cited in the reference (https://proceedings.mlr.press/v119/croce20b/croce20b.pdf). The AC also argued that “The only evidence presented by the authors in this regard is comparing the BPDA attack against a single step white-box gradient based attack”. This is also not true. The authors did not use single-step gradient-based attack at all. In Table 2, N=1 means using one-step gradient optimization for likelihood maximization, instead of one-step gradient-based attack. The authors indeed utilized the evaluation method that the AC regards as sufficient, but it appears there is a factual error in the assessment, as AC mistakenly thought the evaluation as a single-step attack.
>
> In their experiments, they evaluated the simplest case of RDC (N=1) using a white-box exact gradient attack with AutoPGD to assess its robustness. The results indicate a robustness of 69.53%. For comparison, the Backward Pass Differentiable Approximation (BPDA) method achieved a robustness of 69.92% for this case. Based on these findings, the authors present two arguments: (1) BPDA is a sufficient method for evaluating RDC, and (2) The robustness of 69.53% for N=1 is obtained under a sufficiently rigorous evaluation.

---

### Decision · Program_Chairs · 2024-01-16

Reject